# Combining Colchicine and Antiplatelet Therapy to Tackle Atherothrombosis: A Paradigm in Transition?

**DOI:** 10.3390/ijms26031136

**Published:** 2025-01-28

**Authors:** Salvatore Giordano, Marina Camera, Marta Brambilla, Gianmarco Sarto, Luigi Spadafora, Marco Bernardi, Antonio Iaconelli, Domenico D’Amario, Giuseppe Biondi-Zoccai, Alessandra Ida Celia, Elena Tremoli, Giacomo Frati, Dominick J. Angiolillo, Sebastiano Sciarretta, Mattia Galli

**Affiliations:** 1Department of Medical and Surgical Sciences, Division of Cardiology, ‘Magna Graecia’ University, 88100 Catanzaro, Italy; sasigiordano@gmail.com; 2Department of Pharmaceutical Sciences, Università degli Studi di Milano, 20133 Milan, Italy; marina.camera@cardiologicomonzino.it; 3Centro Cardiologico Monzino IRCCS, 20138 Milan, Italy; marta.brambilla@cardiologicomonzino.it; 4Department of Medical-Surgical Sciences and Biotechnologies, Sapienza University of Rome, 04100 Latina, Italy; sarto.gianmarco@gmail.com (G.S.); luigispadafora167@gmail.com (L.S.); marco.bernardi23@gmail.com (M.B.); gbiondizoccai@gmail.com (G.B.-Z.); giacomo.frati@uniroma1.it (G.F.); sebastiano.sciarretta@uniroma1.it (S.S.); 5Department of Cardiovascular Medicine, Fondazione Policlinico Universitario A. Gemelli IRCCS, 00168 Rome, Italy; iaconelli.antonio88@gmail.com; 6Department of Translational Medicine, Division of Cardiology, Università del Piemonte Orientale, AOU Maggiore della Carità, 28100 Novara, Italy; domenico.damario@gmail.com; 7Maria Cecilia Hospital, GVM Care & Research, 48033 Cotignola, Italy; etremoli@gvmnet.it; 8Rheumatology, Department of Clinical Internal, Anestesiological e Cardiovascular Sciences, Sapienza University of Rome, 00161 Rome, Italy; alessandraida.celia@uniroma1.it; 9IRCCS NeuroMed, 86077 Pozzilli, Italy; 10Division of Cardiology, University of Florida College of Medicine, Jacksonville, FL 32209, USA; dominick.angiolillo@jax.ufl.edu

**Keywords:** atherothrombosis, inflammation, colchicine, antiplatelet, thrombo-inflammation

## Abstract

Atherothrombosis, the primary driver of acute cardiovascular (CV) events, is characterized by the activation of three key pathophysiological pathways: platelets, coagulation, and inflammation. Dual antiplatelet therapy (DAPT) is the current standard of care for patients with acute coronary syndrome, providing significant reductions in cardiovascular (CV) events, albeit with an associated increased risk of bleeding. However, the high residual risk of recurrent events among these patients highlights the need for alternative strategies to treat and prevent atherothrombosis. To this extent, several approaches aimed at targeting atherothrombosis have been proposed. Among these, a strategy of dual-pathway inhibition simultaneously targeting platelets, using single or DAPT, and coagulation, using a low-dose anticoagulant such as rivaroxaban 2.5 mg twice daily, has shown to reduce CV events but at the expense of increased bleeding. Targeting inflammatory pathways has the potential to be a highly effective strategy to tackle atherothrombosis without increasing bleeding risk. Several anti-inflammatory agents have been tested in patients with coronary artery disease, but to date only colchicine is approved for secondary prevention on top of standard care, including antiplatelet therapy. However, many aspects of colchicine’s mechanism of action, including its antiplatelet effects and how it synergizes with antiplatelet therapy, remain unclear. In this review, we summarize the available clinical and pre-clinical evidence on the antiplatelet effects of colchicine and its synergistic interactions with antiplatelet therapy, highlighting their potential role in addressing atherothrombosis.

## 1. Background

Despite considerable progress in the treatment and prevention of atherosclerotic cardiovascular diseases (ASCVDs), they continue to be the leading cause of morbidity and mortality globally [1]. In the United States alone, one person dies every 33 s from ASCVDs, with coronary artery disease (CAD) contributing to over 655,000 deaths each year [2]. Atherosclerosis is a chronic and progressive disease of the arteries characterized by the accumulation of fibrofatty lesions, or plaques, within the arterial walls [3]. Atherosclerotic plaque disruption with superimposed thrombosis, known as “atherothrombosis”, is the main actor of acute cardiovascular (CV) events [4,5]. In patients with CAD, this can lead to thrombotic obstruction/occlusion of the coronary artery, resulting in acute coronary syndrome (ACS), or to plaque healing, which contributes to the progression of stable coronary artery syndrome (CCS) [6,7] (Figure 1). Atherothrombosis has been the primary target of multiple pharmacological strategies for the primary and secondary prevention of CAD, aimed at reducing atherosclerotic plaque progression and promoting plaque stabilization, as well as strategies for the acute treatment of ACS, reducing the thrombotic burden responsible for acute artery obstruction/occlusion and consequent microvascular dysfunction [4,6,8]. Given the crucial role of platelets in thrombus formation within the arterial vasculature, antiplatelet agents were one of the initial approaches adopted to tackle atherothrombosis [6,9].

In this regard, aspirin, initially used alone and later in combination with P2Y_12_ inhibitors (e.g., clopidogrel, prasugrel, and ticagrelor) in a strategy known as “dual antiplatelet therapy” (DAPT), has been shown to significantly reduce thrombotic events and now represents the standard of care in patients with ACS [6]. However, this benefit comes at the cost of an increased bleeding [10]. However, the still-significant risk of ischemic recurrences after ACS, that is particularly relevant in high-risk patients, underscores the need for targeting alternative pathways involved in atherosclerotic disease progression and atherothrombosis.

To this extent, the increasing understanding of the pivotal role of coagulation in thrombotic as well as in inflammatory processes involved in the pathogenesis of atherosclerosis and atherothrombosis has represented the rationale for the development of treatment regimens targeting both platelets and coagulation in patients with ASCVD [11,12,13] (Figure 1). This strategy, known as “dual-pathway inhibition” (DPI), combining antiplatelet agents such as a low dose of aspirin and/or a P2Y_12_ inhibitor with a low dose of rivaroxaban (i.e., 2.5 mg twice daily) [13]. DPI showed promising pharmacodynamic effects [14,15,16], resulting in a reduced incidence of major adverse cardiovascular events (MACE) among the spectrum of patients with ASCVD, compared to a strategy of single or DAPT, but was associated with a doubled risk of major bleeding, making it eligible only for a minority of patients presenting with high ischemic and low bleeding risks [17].

In this context, the fact that inflammation plays a fundamental role in both atherogenic and atherothrombotic processes has prompted investigations into the use of anti-inflammatory agents in adjunct to antiplatelet agents to improve outcomes in ASCVD patients without increasing the risk of bleeding. Several anti-inflammatory agents have been tested in patients with ASCVD receiving concomitant antiplatelet therapy [18,19], but only colchicine has garnered sufficient evidence to be recommended by international guidelines for patients with a history of ACS [20].

However, colchicine’s mechanism of action remains incompletely understood. Emerging evidence suggests that colchicine may also exhibit antiplatelet properties, indicating that its CV benefits may at least partially stem from its ability to inhibit platelet activity. These mechanisms may be particularly relevant in clinical practice, especially considering that colchicine is frequently added to antiplatelet therapy in patients with ASCVD. This combination has the potential to influence not only thrombotic but also bleeding events [21].

In this review, we summarize the available clinical and pre-clinical evidence on the antiplatelet effects of colchicine and its synergistic interactions with antiplatelet therapy, highlighting their potential role in addressing atherothrombosis.

## 2. Interplay Between Coagulation, Platelets, and Inflammation in Atherothrombosis

Recent evidence suggests that platelets, coagulation, and inflammatory processes are not three separate and independent entities, but rather part of a complex and interconnected system responsible for multiple physiological and pathophysiological functions, such as the immune response, anti-tumor response, thrombus formation, autoimmune diseases, and the progression and destabilization of atherosclerotic disease, including atherothrombosis [22,23].

A fundamental link between platelets, coagulation, and inflammation is played by thrombin. In fact, according to the more recent “cell-based model of coagulation”, the extrinsic and intrinsic pathways of the coagulation cascade and platelets are sequentially involved and results in thrombin formation. Within this context, it is worth mentioning that activated platelets not only provide the best surface for the assembly of the coagulation factors, but they also express tissue factor (TF), the key activator of the blood coagulation cascade, becoming independent in triggering coagulation [24,25,26]. Thrombin acts through PARs, G protein-coupled receptors (GPCRs) that have been identified in platelets, endothelial cells, leukocytes, and vascular SMCs [12,27]. In humans, four members of the PAR family have been identified. PAR-1 is activated by FXa and thrombin (FIIa), PAR-2 by FXa and the FVIIa-TF complex, while PAR-3 and PAR-4 are activated by FIIa. Activation of PARs exerts significant effects on both platelets and inflammation. On the platelet side, PARs trigger the release of granules containing various chemokines, such as platelet factor 4, interleukin (IL)-1, IL-8, macrophage inflammatory protein, and Regulated upon Activation, Normal T-cell Expressed and Secreted (RANTES). They also mobilize intracellular calcium and promote the expression of adhesion molecules like P-selectin and CD40 ligand, along with prothrombinase-binding phosphatidylserine on the platelet surface. This leads to platelet aggregation and pro-inflammatory effects, including the formation of platelet–leukocyte aggregates [12,27]. Thrombin not only plays a central role in thrombus formation, but also drives a range of pro-inflammatory processes [22,23,28]. These include the recruitment and activation of monocytes and neutrophils, induction of leukocyte adhesion molecules in endothelial cells, release of pro-inflammatory cytokines, and activation of the complement system [22,23,28]. These processes synergistically contribute to both thrombosis and inflammation [13,28].

On the inflammatory side, PARs stimulate endothelial cells, monocytes, and fibroblasts to release pro-inflammatory molecules, including monocyte chemoattractant protein-1, tumor necrosis factor (TNF)α, and interleukins (IL-1β, IL-6, IL-8). They also activate endothelial cells, leading to the expression of adhesion molecules such as P- and E-selectins. Notably, there is a bidirectional relationship between coagulation, platelets, and inflammation, as pro-inflammatory interleukins (e.g., IL-1 and IL-6) can activate platelets and the coagulation cascade through multiple pathways [12,29].

Inflammation influences thrombosis through several other mechanisms, such as endothelial injury, reduction of endogenous anticoagulants, inhibition of fibrinolytic activity, and enhanced pro-coagulant activity via TF-mediated coagulation activation [22,23,30].

Healthy endothelium plays a crucial role in controlling hemostasis. Endothelial cells support natural anticoagulant processes by producing thrombomodulin, which activates protein C and deactivates thrombin. They also express heparan and dermatan sulfates, enhancing the activity of antithrombin III (AT-III) and heparin cofactors. Additionally, endothelial cells promote fibrinolysis by expressing TF pathway inhibitor (TFPI) and producing tPA and uPA. Furthermore, normal endothelium releases nitric oxide (NO) and prostacyclin (PGI2), leading to vasodilation. Inflammation disrupts this balance, increasing procoagulant and antifibrinolytic factors like von Willebrand factor (vWF), Thromboxane A2 (TxA2), plasminogen activator inhibitor type 1 (PAI-1), TF, and cell adhesion molecules, shifting the endothelium toward a proinflammatory and prothrombotic state.

Procoagulant activity is normally regulated by three important anticoagulant pathways: AT-III, the protein C system, and tissue TFPI. Evidence suggests that inflammation may negatively affect anticoagulant pathways such as AT-III and the protein C system. AT-III levels are reduced in chronic inflammation. AT-III protein is digested by elastase produced by neutrophils and consumed by thrombin. Moreover, AT-III activity is reduced due to the consumption of its co-factors (i.e., GAG) from the endothelial surface. Inflammation also disrupts the protein C system. Protein C levels decrease due to reduced synthesis, increased consumption, and degradation by proteolytic enzymes like neutrophil elastase. Additionally, TNF-α and IL-1 downregulate thrombomodulin (TM), leading to reduced protein C activation. Inflammation also raises plasma levels of C4b-binding protein, which can cause a deficiency in free protein S, further promoting a procoagulant state.

As far as it concerns the process of fibrinolysis, pro-inflammatory cytokines (i.e., TNF-α and IL-1) increase the release of fibrinolytic agents, such as tissue plasminogen activator (tPA) and urokinase-type plasminogen activator (uPA) from endothelial cells. This process, however, is counteracted by a sustained increase in PAI-1, resulting in an overall inadequate fibrin removal.

TF is a transmembrane glycoprotein that plays a key role in initiating thrombin generation during inflammation, forming a critical link between inflammation and thrombosis. TF binds FVII leading to its activation into aFVII and catalyzes the conversion of FX into FXa directly or through the activation of FIX, ultimately generating thrombin. TF is primarily located in tissues not directly exposed to blood, such as the adventitial layer of blood vessels, having a primary role in the process of hemostasis [31]. Inflammation increases the synthesis and expression of TF, mainly by endothelial cells, macrophages, and platelets [30,32]. In atherosclerotic plaques, foam cells generate TF, which promotes thrombus formation upon plaque rupture [22,30].

## 3. Colchicine: Pharmacokinetics and Pharmacodynamics

Colchicine is a tricyclic lipophilic alkaloid extracted from *Colchicum autumnale* (commonly known as autumn crocus or meadow saffron) and *Gloriosa superba* (glory lily), where it is found in the corn, seeds, and flowers [33,34]. Known since ancient Egyptian times, colchicine was approved for medical use in the United States by the Food and Drug Administration in 1961 and ranked as the 200th most prescribed medication in the United States in 2018, with nearly three million prescriptions [33,34]. It has a long history of use in treating various medical conditions, including gout, familial Mediterranean fever, Behcet’s disease, and pericarditis.

Colchicine is administered orally in tablet forms of 0.5, 0.6, and, in some regions, 1 mg. It is primarily absorbed in the jejunum and ileum, as dysfunction in these areas is frequently observed in chronic colchicine overdose. The multidrug transporter P-glycoprotein 1 (PGY1) plays a key role in colchicine’s absorption and excretion, affecting its bioavailability, which varies widely from 24% to 88% [35]. Notably, colchicine bioavailability is comparable in the young and elderly [35]. Cytochrome P450 3A4 (CYP3A4) is involved in the catabolism of up to 20% of colchicine, which is transformed into 2 and 3 demethylcolchicine, inactive metabolites. Most of the compound is eliminated via bile but also via gastrointestinal tract cell lining turnover, and roughly 10–20% is excreted by urine. Notably, alteration of PGY-1 functioning may affect CYP3A4 and renal disposition of colchicine [35].

After administration, the peak plasma concentration of colchicine occurs within 0.5–2.0 h. However, most of the drug undergoes enterohepatic recirculation, leading to a second plasma peak within 6 h [36]. In plasma, approximately 40% of colchicine conjugates with plasma proteins. Colchicine has a high volume of distribution (7 L/kg), meaning it accumulates into tissue and cells, including leukocytes, where colchicine levels peak at a mean of 47 h and are persistently present even 10 days after drug discontinuation [37]. The elimination half-life of colchicine is 20–40 h, which could be prolonged in case of renal failure or hepatic cirrhosis [37,38].

Colchicine interacts with multiple drugs metabolized by CYP3A4 and P-glycoprotein. PGY-1 metabolized drugs commonly used in ASCVD patients which can interact with colchicine include statins and calcium channel blockers.However, particular toxicity has been described with co-administration with cyclosporine [39]. Cyclosporine modulates the PGY-1, and the co-administration potentiates colchicine neuromyopathy and cyclosporine nephropathy with a declining glomerular filtration rate [40]. Notably, interactions with CYP3A4 inhibitors like clarithromycin, fluoxetine, indinavir, ketaconazole, non-dihydropyridine calcium channel blockers, nefazodone and protease inhibitors, among others, can impair colchicine metabolism, potentially resulting in fatal outcomes when combined with clarithromycin [41]. Given these complexities, dose adjustments of colchicine are imperative when co-administered with these agents, as outlined in the prescribing information. Such adjustments are critical to mitigate adverse events, particularly in patients with renal impairment.

## 4. Colchicine: Mechanism of Action

The pharmacological effect of colchicine is not fully understood. It is believed that the main mechanism of action is secondary to its interaction with tubulin heterodimers, which modulates microtubules activity. At lower doses, colchicine impedes further microtubule growth, while at higher doses, it promotes microtubules depolymerization [42]. Microtubules serve as dynamic filaments crucial for cytoskeletal architecture and function and play a pivotal role in an array of cellular activities including cell division, signal transduction, gene expression regulation, cellular migration, and secretion processes [34].

There are numerous mechanisms through which colchicine exerts its anti-inflammatory effects [43]. Colchicine highly concentrates in innate immunity cells, particularly in granulocytes, where it inhibits several functions, including adhesion, motility, and degranulation, fostering inhibition of leukocytes chemotaxis [41]. Colchicine modulates adhesion molecules expression on endothelial, particularly reducing E-selectin-mediated leukocyte adhesion to endothelial cells [44]. Furthermore, colchicine suppresses the activation of caspase-1, the enzymatic component of the inflammasome [45], which in turn hampers the activation of IL-1β yielding to the reduction of TNF-α and IL-6. Finally, colchicine reduces superoxide production by neutrophils [5].

Activated platelets undergo a process of dynamic shape change and granule secretion, which require dramatic alteration of the cytoskeleton [46]. Resting platelets have a discoid shape and, when activated, they flatten out to form extensions called lamellipodia and filopodia to ultimately form platelet aggregates. The cytoskeletal apparatus of platelets consists of a network of microtubules, actin filaments, and myosin. Microtubules (25 nm) are made of tubulin monomers and are organized in a circumferential loop known as “marginal ring”, right under the plasma membrane, being responsible for the classical discoid shape of the platelets. Actin matrix, on the other hand, is spread around the platelet and is fundamental for the process of the cytoskeletal rearrangement during platelet activation [46].

In vitro and ex vivo studies have shown colchicine to exert antiplatelet effects, primarily mediated by the inhibition of key proteins involved in cytoskeletal rearrangement [47,48,49,50,51]. In particular, immunohistochemical analyses revealed that colchicine inactivates proteins involved in the regulation of microtubules Myosin Phosphatase Targeting subunit (MYPT) and LIM domain kinase 1(LIMK1), ultimately interfering with cofilin activity, inducing microtubules depolymerization and cytoskeleton disarrangement [48,49,50,51]. A detailed description of the in vitro and in vivo studies exploring the antiplatelet effects of colchicine is described below.

## 5. Possible Discomforts and Risks of Colchicine Therapy

In clinical trials, the most frequently reported adverse effects associated with colchicine were gastrointestinal symptoms, including diarrhea, flatulence, and abdominal discomfort, affecting approximately 15% of patients [52,53,54,55,56,57]. However, these rates were comparable to those observed in placebo groups. Hospitalizations for infections, pneumonia, or gastrointestinal conditions, as well as the diagnosis of cancer, were also similar between colchicine-treated patients and those receiving placebo [52,53,54,55,56,57]. Furthermore, no significant differences were noted in the incidence of neutropenia, myotoxicity/myalgia, or dysesthesia. One study did report an increased risk of pneumonia associated with colchicine use, but no higher risk of overall infections or septic shock was identified [52,53,54,55,56,57]. Although low-dose colchicine has demonstrated a favorable safety profile in patients with CV disease, it remains associated with potential side effects across various systems: digestive: abdominal pain, cramping, diarrhea, nausea, vomiting, and lactose intolerance; neurological: sensory–motor neuropathy; dermatological: alopecia, maculopapular rash, purpura, and generalized rash; hematological: leukopenia, granulocytopenia, thrombocytopenia, pancytopenia, and aplastic anemia; hepatobiliary: elevated liver enzymes (aspartate aminotransferase, alanine transaminase); musculoskeletal: myopathy, elevated creatine phosphokinase levels, myotonia, muscle weakness, muscle pain, and rhabdomyolysis; reproductive: azoospermia and oligospermia. This broad spectrum of potential side effects underscores the importance of careful patient selection and monitoring during colchicine therapy, particularly in populations at higher risk for adverse outcomes.

Colchicine is used to prevent attacks in patients with familial Mediterranean fever; however, not all the patients clinically respond to the therapy. Those patients are known as colchicine non-responders, and a study has shown that this condition is associated with a reduced concentration of colchicine in mononuclear cells (concentration of colchicine in ng/10^9^ cells in non-responders vs. responders: 102 ± 67 vs. 234 ± 169, *p* = 0.001). This effect can be attributed to a potential genetic effect [58]. In particular, the presence of an ABCB1 3435 C allele is associated with an increased risk of colchicine resistance and the ABCB1 3435 T-genotypes with an increased response. Notably, whether colchicine non-responsiveness may affect clinical benefit in terms of cardiovascular outcomes as compared to colchicine responders remains to be explored.

## 6. Antiplatelet Effects of Colchicine

We searched for in vitro and in vivo studies exploring the antiplatelet effects of colchicine in humans. Only studies in English were included. Table 1 summarizes the main characteristics and findings of included studies.

### 6.1. In Vitro Studies

An in vitro study by Shah et al. has shown that the addition of colchicine to the whole blood of healthy subjects decreased platelet activation in terms of the formation of monocyte–platelet and granulocyte–platelet hetero-aggregates [49]. This effect was already observed at concentrations comparable with the therapeutic concentration achieved in vivo (which is ~0.015 μM after ingestion of 1.8 mg PO load of colchicine over 1 h).

More recently, Brambilla et al. showed that incubation of the whole blood of healthy subjects with colchicine significantly reduces in a concentration-dependent manner (0.020 μM to 100 μM) the percentage of activated glycoprotein (aGP) IIbIIIa^pos^-platelets (measured by flow-cytometry) induced by adenosine Diphosphate (ADP) stimulation. The expression of P-selectin, one of the key proteins involved in platelet–leukocyte aggregate formation, was also downregulated resulting in a lower percentage of P-selectin^pos^-platelets with the highest concentration of colchicine used [59].

When the effect of colchicine was assessed in platelet-rich plasma (PRP), the drug significantly reduced platelet aggregation assessed by Light Transmission Aggregometry (LTA) at supratherapeutic concentrations [49].

This finding was similarly reported by Cirillo et al. showing that the addition of colchicine (10 μM) to the PRP of patients on a chronic clopidogrel-based DAPT regimen further modulates platelet reactivity [51]. In this study, 28 clopidogrel responders and seven clopidogrel non-responders were included. The level of platelet aggregation was evaluated by LTA and expressed as maximal aggregation in PRP after activation at baseline, 30, 60, and 90 min, in response to ADP (20 μM) and Thrombin Receptor Activating Peptide (TRAP 25 μM) [51]. When platelets from all patients were treated with colchicine, a significant reduction of TRAP-induced aggregation as compared to platelets not treated with colchicine was observed [51]. Interestingly, in clopidogrel responders, ADP-induced maximal aggregation in the presence of colchicine was similar to that obtained without the drug [51]. On the other hand, the addition of colchicine to the platelets of the clopidogrel non-responders group significantly reduced ADP-induced platelet aggregation at 30 (80 ± 3% vs. 22 ± 12%), 60 (79 ± 4% vs. 19 ± 11%), and 90 min (75 ± 4% vs. 21 ± 8%), *p* < 0.05 [51].

Although it can be argued that the concentration of colchicine used in these in vitro studies to modulate platelet aggregation in PRP was significantly higher than the therapeutic concentration, it should be considered that the experimental setup with a limited colchicine pre-incubation time, as well as preanalytical variables(such as preparation of PRP) could account for the fact that higher than therapeutic drug concentrations are required to see a biological effect. Increasing evidence suggests that colchicine may influence platelet activation leading to thrombus formation through mechanisms beyond the activation of GPIIbIIIa required to sustain platelet aggregation. TF, the key protein triggering the coagulation cascade, is also expressed by a subset of platelets [24,25]. ADP induces a seven-fold increase in the number of TF^pos^-platelets, a process mediated exclusively via the P2Y_12_ receptor [59]. Evidence suggests that TF is stored in the open canalicular system (OCS) of platelets and displayed on the cell surface via externalization upon platelet activation [59]. Colchicine hampers the process of OCS externalization, reducing agonist-induced TF expression [59]. In vitro, the incubation of the whole blood of healthy subjects with increasing concentrations of colchicine (0.020 μM to 100 μM, 15 min) significantly reduces the percentage of TF^pos^-platelets induced by ADP thus affecting the platelet pro-thrombotic potential [59].

All together, these in vitro data highlight that when the effects of colchicine on platelet activation have been evaluated in whole blood, they suggest that different pathways can be targeted, spanning from the classical platelet activation markers, such as GPIIbIIIa and platelet–leukocyte aggregates, to the expression of TF, relevant for the platelet prothrombotic phenotype.

### 6.2. In Vivo Studies

In line with in vitro experiments performed on whole blood, in vivo studies on healthy subjects confirmed that colchicine significantly reduces monocyte- and neutrophil–platelet aggregate levels, as well as P-selectin expression, assessed by whole blood flow cytometry 2 h after administering a 1.8 mg dose (although not at 24 h) [49]. The platelet surface expression of aGPIIbIIIa was also significantly reduced at both the assessed timepoints. By contrast, no significant effect of colchicine on platelet aggregation in PRP in response to 1 μM ADP and 0.4 μM epinephrine was observed [49].

Similar results were found by a pilot study that tested the effects of a daily administration of 1 mg of colchicine on platelet reactivity and on inflammatory markers [60]. This was a randomized, placebo-controlled study conducted on 80 patients with ACS or acute ischemic stroke [60]. At 30 days, colchicine did not affect platelet function, assessed byLTA following stimulation with ADP (5 μmol), arachidonic acid (0.5 mmol), collagen (1 μg/mL and 5 μg/mL) (*p* = 0.86, *p* = 0.64, *p* = 0.76, *p* = 0.20, respectively), and urine dehydrothromboxane B_2_ (*p* = 0.54) [60].

The link between the degree of P2Y_12_ inhibition and TF expression in platelets was also confirmed in vivo. Clopidogrel responders showed a significantly lower percentage of TF^pos^-platelets as compared to poor responders (*p* < 0.0001) or patients not on P2Y_12_ inhibitors (*p* < 0.0001) [59]. Furthermore, prasugrel and ticagrelor users showed a reduced percentage of TF^pos^-platelets compared to clopidogrel responders. However, the P2Y_12_ inhibitor concentration needed to inhibit cell surface TF expression was much higher than that needed to inhibit the expression of the classical platelet activation markers such as P-selectin, aGPIIbIIIa, and VASP [59].

Therefore, available evidence suggests that the addition of colchicine to P2Y_12_ inhibitors could further hamper platelet activation by reducing the expression of other pro-thrombotic molecules, including TF. This could be useful especially in patients who do not respond sufficiently to clopidogrel.

The Mono Antiplatelet and Colchicine Therapy (MACT) trial was a single-arm, open-label, proof-of-concept pilot trial that tested the feasibility of a P2Y_12_ inhibitor monotherapy combined with colchicine strategy immediately after PCI in patients with ACS [61]. A total of 200 patients were included in the study and received ticagrelor or prasugrel (at the physician’s discretion) plus colchicine 0.6 mg/day and were followed up for 3 months. Patients received aspirin only on the day of PCI [61]. The primary outcome was stent thrombosis, which occurred in two patients (1.0%), and secondary outcomes included platelet reactivity before discharge (0.5% had HPR) and reduction of high-sensitivity C-reactive protein (hsCRP) levels over 1 month (decreased from 6.1 mg/L to 0.6 mg/L). Overall, the MACT approach appeared to be safe and had favorable platelet function and inflammatory profiles [61].

## 7. Current Evidence and Future Perspectives

Colchicine has shown several in vitro and in vivo effects that support the rationale for its use in the treatment and prevention of atherothrombosis in patients with atherosclerotic cardiovascular disease, particularly those with coronary artery disease. The association of colchicine and a single or dual antiplatelet therapy, particularly a P2Y_12_ inhibitor, may represent a new frontier to tackle atherothrombosis with a more favorable safety profile compared to other strategies focusing on platelets and coagulation pathways, such as DAPT or DPI strategies (Figure 1).

Further research is needed to elucidate the pathophysiological mechanisms underlying the combined use of colchicine and antiplatelet agents. This includes exploring the interactions between colchicine’s anti-inflammatory properties and the antithrombotic effects of antiplatelet therapies, as well as their collective impact on platelet function, vascular inflammation, and thrombotic pathways. Additionally, robust clinical studies are essential to provide comprehensive evidence supporting the safety, efficacy, and optimal therapeutic regimens for this combination strategy. Indeed, thus far, clinical studies evaluating the effects of colchicine in patients with CAD have yielded mixed results (Table 2). Five pivotal randomized controlled trials (RCTs) have investigated whether the addition of colchicine on top of optimal medical therapy could improve the outcomes for the primary and secondary prevention of patients affected by ASCVD [52,53,54,55,56]. Overall, these trials found that colchicine reduced MACE by 25%, myocardial infarction by 22%, stroke by 46%, and coronary revascularization by 23% in patients with both ACS and CCS [62]. Although some concern on the possible higher rate of non-CV death in the colchicine vs. placebo group was raised, a subsequent sub-study of the Low-Dose Colchicine 2 (LoDoCo2) trial found no significant association with any specific causes of death [63,64].

Evidence from these RCTs led the 2023 ACS guidelines from the European Society of Cardiology (ESC) to recommend the use of low-dose colchicine (0.5 mg once a day) with a class IIb, level of evidence (LoE) A in ACS patients, particularly if other risk factors are insufficiently controlled or if recurrent CV disease events occur under optimal therapy [66]. More recently, the 2024 CCS guidelines from the ESC recommended the use of low-dose colchicine with a class IIa, LoE A in CCS patients with atherosclerotic CAD to reduce MI, stroke, and the need for revascularization [20].

However, these guidelines were released before the publication of the largest CLEAR SYNERGY (OASIS 9) trial [57]. CLEAR SYNERGY enrolled 7062 patients at 104 centers in 14 countries and found that, after 5 years of follow-up, daily treatment with colchicine 0.5 mg daily in patients who suffered from an MI and underwent PCI did not reduce MACE compared with placebo [57]. Non-univocal results also came from other clinical studies testing the effects of colchicine in other specific clinical settings. In brief, conflicting results on the effectiveness of colchicine in reducing infarct size in patients with acute MI [65,67,72] and improving outcomes after PCI have been reported [56,68,69]. On the other hand, colchicine has shown to promote atherosclerotic plaque stabilization [70,71] and modulate clonal hematopoiesis (CH), a new emerging and independent cardiovascular risk factor [73].

The conflicting clinical outcomes observed in large trials highlight, on one hand, the pressing need for more robust and conclusive clinical studies to reinforce guideline recommendations regarding its use. On the other hand, they emphasize the necessity for a deeper understanding of the pathophysiological mechanisms driving colchicine’s effects, particularly in combination with antiplatelet therapy.

One possible hypothesis is that only specific subsets of patients benefit from colchicine treatment. If this is the case, patient selection becomes crucial for implementing precision medicine approaches. From this perspective, pathophysiological studies aimed at enhancing our understanding of colchicine’s mechanisms of action and its interactions with other medications, especially antiplatelet agents, are of the utmost importance. These investigations are essential not only to resolve uncertainties regarding efficacy but also to optimize the safe and targeted use of colchicine in clinical practice. Finally, evidence comparing the safety and efficacy of a strategy combining colchicine with antiplatelet agents versus DAPT or DPI, alongside studies aimed at identifying the specific patient profiles, most likely to benefit from dual targeting of inflammatory and platelet pathways rather than focusing solely on thrombotic pathways, could pave the way for personalized treatment approaches and better clinical outcomes.

## Figures and Tables

**Figure 1 ijms-26-01136-f001:**
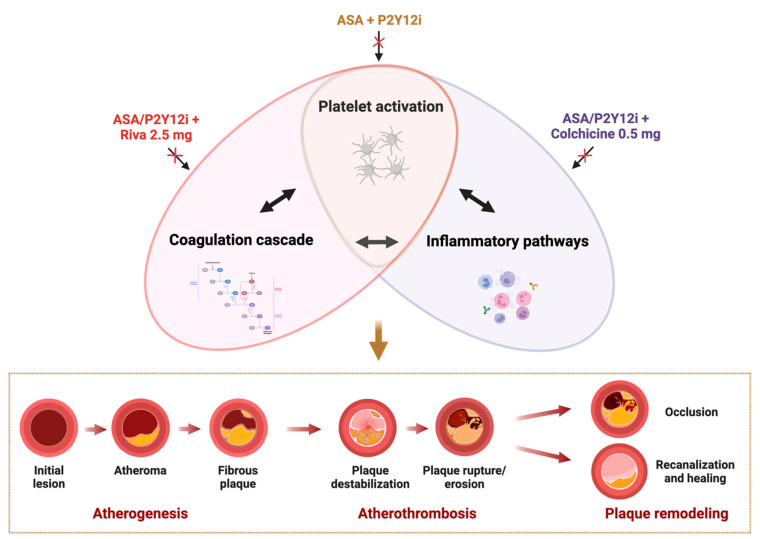
**Strategies for Tackling Atherothrombosis in Patients with Atherosclerotic Cardiovascular Disease.** Emerging evidence underscores the complex interplay between platelets, coagulation, and inflammation in the development of thrombus formation and atherothrombosis. Current therapeutic strategies aim to reduce cardiovascular events by disrupting these pathways that are responsible for atherogenesis and atherothrombosis. Dual antiplatelet therapy, combining aspirin and a P2Y_12_ inhibitor, primarily targets platelet reactivity to prevent thrombotic complications. A more comprehensive approach involves the simultaneous inhibition of platelets and coagulation using aspirin or a P2Y_12_ inhibitor in combination with low-dose rivaroxaban (2.5 mg twice daily). However, because both platelets and coagulation factors are integral to normal hemostasis, these therapies reduce thrombotic events but at the expense of increased risk of bleeding. An alternative approach focuses on combining aspirin or a P2Y_12_ inhibitor with colchicine to target both platelet activity and inflammatory pathways. By addressing inflammation, a key driver of atherothrombosis, this strategy has the potential to reduce thrombotic risk while minimizing the bleeding complications typically associated with intense antithrombotic therapies. Abbreviations: ASA: aspirin, P2Y12i: P2Y_12_ inhibitors; Riva: rivaroxaban.

**Table 1 ijms-26-01136-t001:** Antiplatelet effects of colchicine.

In Vitro Studies	Study Design	Colchicine Dose	Clinical Setting	Sample Size	Main Findings
Shah et al., 2016 [48]	Addition of colchicine to PRP (for 30 min) and whole blood (for 5 min) was followed by assessment of platelet activity and adhesion via LTA and flow cytometer	0.015, 0.15, 1.5, 15, 150, 1500, 15,000 μM	Healthy adults	*n* = 10	Addition of colchicine: PRP significantly decreased platelet–platelet aggregation at >1.5 μM (maximal platelet aggregation 91% vs. 84%, *p* < 0.05) Whole blood decreased MPA (74.5% vs. 51.1%, *p* = 0.04) and NPA (40.6% vs. 26.4%, *p* = 0.04) at 0.015 μM
Brambilla et al., 2023 [58]	Whole blood was incubated with colchicine and then the expression of platelet-associated TF, P-selectin, and GPIIbIIIa was measured by flow cytometry upon stimulation with ADP	20 nM, 100 nm, 1 µM, 10 µM, 100 µM	Healthy adults	*n* = 10	Colchicine reduced in a concentration-dependent manner the following: GPIIbIIIa membrane expression (baseline vs. 100 nM, *p* = 0.022; baseline vs. 1 µM, *p* < 0.0001; baseline vs. 10 µM, *p* = 0.0003; baseline vs. 100 µM, *p* = 0.0049) ADP-induced TF externalization (baseline vs. 1 µM, *p* = 0.026; baseline vs. 10 µM, *p* = 0.001; baseline vs. 100 µM, *p* = 0.0002) At higher concentrations, P-selectin expression (baseline vs. 100 µM, *p* = 0.022)
Cirillo et al., 2020 [50]	PRP was pre-incubated with colchicine before being stimulated with ADP or TRAP. PRP not colchicine preincubated served as controls. The level of platelet aggregation was then evaluated by LTA at 30, 60, and 90 min	10 μM	Patients on DAPT with clopidogrel	*n* = 35 (28 clopidogrel responders and seven clopidogrel non-responders)	Colchicine: Reduced TRAP-induced platelet aggregation in both clopidogrel responders (22 ± 7%; 19 ± 4%; 15 ± 1% [LTA-Platelet aggregation], *p* < 0.05) and non-responders (20 ± 9%; 24 ± 8%; 22 ± 1%, *p* < 0.05) as compared to TRAP-stimulated platelets but not preincubated with colchicine Inhibited platelet aggregation in clopidogrel non-responders in which ADP still caused activation despite DAPT (22 ± 12%, 19 ± 11%, 21 ± 8%, *p* < 0.05)
Shah et al., 2016 [48]	Administration of a 1.8 mg oral colchicine loading dose over one hour. Subsequent blood samples were drawn 2 and 24 h after completion of the loading dose; platelet activity and adhesion were then assessed via LTA, flow cytometer, and fluorescence microscope	1.8 mg over one hour	Healthy adults	*n* = 10	Colchicine Did not have a significant effect on platelet aggregation in response to 1 μM ADP and 0.4 μM epinephrine (maximal platelet aggregation at baseline 84.4% vs. 2 h 84.6% vs. 24 h 87.5%; baseline vs. 2 h, *p* = 0.21; baseline vs. 24 h, *p* = 0.37) Significantly decreased MPA at 2 h but not at 24 h (baseline 27.8% vs. 2 h 22% vs. 24 h 35.6%; baseline vs. 2 h, *p* = 0.051; baseline vs. 24 h, *p* = 0.58) Significantly decreased NPA at 2 h but not at 24 h (baseline 18.9% vs. 2 h 14.7% vs. 24 h 21.8%; baseline vs. 2 h, *p* = 0.013; baseline vs. 24 h, *p* = 0.39) Reduced the platelet surface expression of PAC-1 at 2 h and 24 h (baseline 369.5 [mean fluorescence intensity] vs. 2 h 332.5 vs. 24 h 342.4; baseline vs. 2 h, *p* = 0.017; baseline vs. 24 h, *p* = 0.005) and of P-selectin at 2 h but not at 24 h (baseline 350.6 vs. 2 h 279.4 vs. 24 h 311.8; baseline vs. 2 h, *p* = 0.03; baseline vs. 24 h, *p* = 0.44)
Raju et al., 2012 [59]	Pilot randomized controlled trial comparing the effect of daily colchicine administration with placebo on hs-CRP levels and platelet function by turbidimetric platelet aggregometry	1 mg/day for 30 days	Patients with ACS or acute ischemic stroke	*n* = 80	Colchicine Did not significantly reduce absolute hs-CRP at 30 days (median 1.0 mg/L vs. 1.5 mg/L, *p* = 0.22) Did not affect platelet function assessed using platelet aggregation with ADP (5 μmol), arachidonic acid (0.5 mmol), collagen (1 μg/mL), collagen (5 μg/mL), and urine dehydrothromboxane B2, (*p* = 0.86, *p* = 0.64, *p* = 0.76, *p* = 0.20, respectively)
Lee et al., 2023 [60]	Proof-of-concept pilot trial investigating the feasibility of ticagrelor or prasugrel P2Y_12_ inhibitor monotherapy combined with colchicine immediately after PCI in patients with ACS	0.6 mg daily	ACS patients treated with drug-eluting stents	*n* = 200	In ACS patients undergoing PCI, discontinuing aspirin therapy and administering low-dose colchicine on the day after PCI in addition to ticagrelor or prasugrel is associated with the following: Low incidence of stent thrombosis (1.0% at 3 months); Major bleeding is rare, (with a 3-month incidence of 0.5%); High platelet reactivity at discharge is low (0.5%); Inflammatory levels were rapidly reduced within 1 month as shown by a significant decrease in hs-CRP levels (from 6.1 mg/L at 24 h after PCI to 0.6 mg/L at 1 month, *p* < 0.001)

Abbreviations: PRP = Platelet-Rich Plasma; LTA = Light Transmission Aggregometry; MPA = Monocyte Platelet Aggregation; NPA = Neutrophil Platelet Aggregation; TF = Tissue Factor; GPIIbIIIa = Glicoprotein IIbIIIa; ADP = Adenosine Diphosphate; TRAP = Thrombin Receptor Activating Peptide; DAPT = Dual Antiplatelet Therapy; hs-CRP = High-Sensitivity C-reactive Protein; ACS = Acute Coronary Syndrome; PCI = Percutaneous Coronary Intervention.

**Table 2 ijms-26-01136-t002:** Main characteristics and findings of clinical studies exploring the effectiveness of colchicine in ASCVD patients.

	Study Design	Outcomes	Colchicine Dose	Sample Size	Follow-Up	Main Findings
Nidorf et al., 2013 [51]	Randomized, observer-blinded trial. CCS patients were assigned to colchicine or no colchicine	Primary composite: ACS, out of hospital cardiac arrest, or non-cardioembolic stroke Secondary: individual components of the primary outcome and the components of ACS unrelated to stent disease	0.5 mg/day	Colchicine = 282 Controls = 250	3 years	Colchicine reduced the primary outcome (5.3% vs. 16.0%, HR 0.33, 0.18–0.59, *p* < 0.001) compared with placebo Colchicine reduced the incidence of ACS (4.6% vs. 13.6%, HR 0.33, 0.18–0.63, *p* < 0.001) and non-cardioembolic stroke (0.35% vs. 1.6%, HR 0.23, 0.03–2.03, *p* = 0.011) compared with placebo In a pre-specified secondary on-treatment analysis that excluded 32 patients assigned to colchicine who withdrew within 30 days due to intestinal intolerance and a further seven patients (2%) who did not start treatment, the primary outcome occurred less frequently with colchicine compared to placebo (4.5% vs. 16.0%, HR 0.29, 0.15–0.56, *p* < 0.001)
Nidorf et al., 2020 [52]	Randomized, controlled, double-blind trial. CCS participants were assigned to receive either colchicine or placebo	Primary composite: cardiovascular death, spontaneous (nonprocedural) MI, ischemic stroke, or ischemia-driven coronary revascularization. Secondary composite: cardiovascular death, spontaneous MI, or ischemic stroke	0.5 mg/day	Colchicine = 2762 Controls = 2760	28.6 months	Colchicine reduced the primary outcome compared with placebo (6.8% vs. 9.6%, HR 0.69, 0.57–0.83, *p* < 0.001) Colchicine reduced the key secondary endpoint (4.2% vs. 5.7%, HR 0.72, 0.57–0.92, *p* = 0.007) compared with placebo
Tardif et al., 2019 [53]	Randomized, double-blind trial involving patients recruited within 30 days after an MI. Patients were randomly assigned to receive either low-dose colchicine or placebo	Primary composite: death from cardiovascular causes, resuscitated cardiac arrest, MI, stroke, or urgent hospitalization for angina leading to coronary revascularization Secondary: consisted of the components of the primary end point; a composite of death from cardiovascular causes, resuscitated cardiac arrest, myocardial infarction, or stroke; and total mortality in time-to-event analyses	0.5 mg/day	Colchicine = 2366 Placebo = 2379	22.6 months	Colchicine significantly reduced the primary endpoint (5.5% vs. 7.1%, HR 0.77, 0.61–0.96, *p* = 0.02) compared to placebo Colchicine reduced stroke (HR 0.26, 0.10–0.70), and urgent hospitalization for angina leading to coronary revascularization (HR 0.50, 0.31–0.81), compared to placebo No difference in the secondary composite endpoints between groups (4.7% vs. 5.5%) No difference in total mortality between groups (1.8% vs. 1.8%) No difference in the incidence of diarrhea between groups (9.7% vs. 8.9%, *p* = 0.35) Increased risk of pneumonia with colchicine compared to placebo (0.9% vs. 0.4%, *p* = 0.03)
Tong et al., 2020 [54]	Multicenter, randomized, double-blind, placebo-controlled trial. Patients who presented with ACS and had evidence of coronary artery disease on coronary angiography managed with either PCI or medical therapy were assigned to receive either colchicine or placebo	Primary composite: all-cause mortality, ACS, ischemia-driven (unplanned) urgent revascularization, and non-cardioembolic ischemic stroke in a time to event analysis	0.5 mg twice daily for the first month, then 0.5 mg daily for 11 months	Colchicine = 396 Placebo = 399	12 months	No difference in the primary outcome between groups (24 vs. 38 events, *p* = 0.09) Increase in total death (8 vs. 1, *p* = 0.017) and non-CV death (5 vs. 0, *p* = 0.024) with colchicine No difference in adverse effects between groups (23.0% vs. 24.3%). Most of them were gastrointestinal symptoms (23.0% vs. 20.8%)
Deftereos et al., 2013 [55]	Double-blind, prospective, placebo-controlled study. Diabetic patients with contraindication to a drug-eluting stent, undergoing PCI with a BMS, were randomized to colchicine or placebo. Angiography and intravascular ultrasound was performed 6 months after the index PCI	Primary: Angiographic and IVUS restenosis Secondary: angiographic and IVUS parameters of lumen loss and in-stent neointimal hyperplasia	0.5 mg twice daily	Colchicine = 100 Placebo = 110	6 months	Colchicine significantly reduced angiographic restenosis (16% vs. 33%, *p* = 0.007) and IVUS restenosis (24% vs. 43%, *p* = 0.006) Colchicine significantly reduced minimum lumen diameter (2.8 mm (2.2–3.1) vs. 2.3 mm (1.3–2.9), *p* < 0.01) Gastrointestinal symptoms (diarrhea and nausea) were the most common adverse events in the colchicine group (16% vs. 7%, *p* = 0.058)
Opstal et al., 2023 [63]	Randomized, parallel, double-blind trial that evaluated the effect of adding colchicine or placebo in patients with chronic coronary disease	Cause-specific mortality data were analyzed, stratified by treatment status	0.5 mg once daily	Colchicine = 2762 Placebo = 2760	29 months	No difference in CV death between groups (0.7% vs. 0.9%) No difference in non-CV death between groups (1.9% vs. 1.3%)
Jolly et al., 2024 [56]	Multicenter trial with a two-by-two factorial design randomly assigning patients who had myocardial infarction to receive either colchicine or placebo and either spironolactone or placebo.	Primary composite: death from cardiovascular causes, recurrent MI, stroke, or unplanned ischemia-driven coronary revascularization	For the first 90 days: patients weighing >70 kg 0.5 mg twice daily, if <70 kg 0.5 mg daily. After 90 days, 0.5 mg daily for all patients	Colchicine = 3528 Placebo = 3534	2.98 years	No difference in the primary outcome between groups (9.1% vs. 9.3%, *p* = 0.93) Colchicine increased the incidence of diarrhea (10.2% vs. 6.6%, *p* < 0.001)
Mewton et al., 2021 [65]	Double-blind multicenter trial. Patients admitted for a first episode of STEMI referred for PCI were randomized to receive colchicine or placebo from admission to day 5. Patients underwent a cardiac magnetic resonance at 5 days and at 30 days	Primary: reduction of IS at 5 days. Secondary: LV end-diastolic volume change at 3 months and IS at 3 months	2 mg loading dose followed by 0.5 mg twice a day for 5 days	Colchicine = 101 Placebo = 91	3 months	At 5 days, IS did not differ between the colchicine and placebo groups (26 g vs. 28.4 g of LV mass, *p* = 0.87) At 3 months follow-up, there was no significant difference in LV remodeling (colchicine + 2.4% vs. −1.1% change in LV end-diastolic volume, *p* = 0.49) Infarct size at 3 months was also not significantly different between the colchicine and placebo groups (17 g vs. 18 g, *p* = 0.92) The incidence of gastrointestinal adverse events during the treatment period was greater with colchicine than with placebo (34% vs. 11%, *p* = 0.0002)
Bouleti et al., 2024 [66]	Follow-up analysis of the COVERT-MI study on prespecified secondary clinical endpoints	Primary composite: all-cause death, ACS, heart failure events, ischemic strokes, sustained ventricular arrhythmias, and acute kidney injury	2 mg loading dose followed by 0.5 mg twice a day for 5 days	Colchicine = 101 Placebo = 91	1 year	No significant difference regarding the number of MACEs between groups No differences in the occurrence of ischemic strokes: colchicine 3% vs. placebo 2.2% (*p* = 0.99)
Deftereos et al., 2015 [55]	Prospective, double-blinded, placebo-controlled study. Patients presenting with STEMI ≤12 h from pain onset (treated with PCI) were randomly assigned to colchicine or placebo for 5 days. A subset of patients underwent cardiac MRI 6 to 9 days after the index STEMI (MRI subgroup)	Primary: area under the curve of CK-MB fraction concentration over 72 h after admission Secondary: Maximal high-sensitivity troponin T measure during the same time-period. In MRI subgroup, absolute MI volume, determined by LGE, was the primary outcome measure	Loading dose of 2 mg (1.5 mg initially followed by 0.5 mg 1 h later) and continuing with 0.5 mg twice daily	Colchicine = 77 Placebo = 74 MRI subgroup = 60	5 days, until 9 days for MRI subgroup	The area under the creatine kinase–myocardial brain fraction curve was for colchicine group 3144 ng h^−1^·mL^−1^ vs. placebo group: 6184 ng·h^−1^·mL^−1^, *p* < 0.001 Median maximum high-sensitivity troponin T was 19,763 pg/mL and 45,550 pg/mL in the colchicine and control group, respectively (*p* = 0.001) Indexed MRI-late gadolinium enhancement-defined infarct size was 18.3 mL/1.73 m^2^ in the colchicine vs. 23.2 mL/1.73 m^2^ in control group (*p* = 0.019) The relative infarct size (as a proportion to left ventricular myocardial volume) was 13.0% in the colchicine and 19.8%, in the control group (*p* = 0.034)
Shah et al., 2020 [67]	Randomized, double-blind, placebo-controlled trial. Subjects referred for possible PCI were randomized to acute pre-procedural oral administration of colchicine or placebo	Primary: PCI-related myocardial injury according to the Universal Definition Secondary: Occurrence of 30-day MACEs (earliest occurrence of death from any cause, nonfatal MI, or target vessel revascularization) PCI-related MI as defined by the SCAI (76)	1.2 mg 1 to 2 h before coronary angiography, followed by colchicine; 0.6 mg 1 h later or immediately pre-procedure	Colchicine = 366 Placebo = 348 Colchicine + PCI = 206 Placebo + PCI = 194	30 days	Primary outcome did not differ between colchicine and placebo groups (57.3% vs. 64.2%, *p* = 0.19) Secondary composite outcome: colchicine 11.7% vs. placebo 12.9% (*p* = 0.82) PCI-related MI defined by the SCAI (colchicine 2.9% vs. placebo 4.7%, *p* = 0.49)
Cole et al., 2021 [68]	Randomized pilot trial. Patients undergoing PCI for stable angina or NSTEMI were randomized to oral colchicine or placebo, 6 to 24 h pre-procedure	Primary: periprocedural myocardial infarction	1 mg followed by 0.5 mg 1 h later	Colchicine = 36 Placebo = 39	24 h	No patients developed periprocedural myocardial infarction in either group Colchicine significantly reduced major periprocedural myocardial injury: colchicine 31% vs. placebo 54%, *p* = 0.04 Colchicine significantly reduced minor periprocedural myocardial injury: colchicine 58% vs. placebo 85%, *p* = 0.01
Yu et al., 2024 [69]	Prospective, single-center, randomized, double-blind clinical trial. Patients with ACS with lipid-rich plaque detected by optical coherence tomography were included. The subjects were randomly assigned to receive either colchicine or placebo	Primary: Change in the minimal fibrous cap thickness from baseline to the 12-month follow-up	0.5 mg once daily	Colchicine = 52 Placebo = 52	12 months	Colchicine increased the minimal fibrous cap thickness (51.9 μm vs. 87.2 μm, *p* = 0.006), and reduced average lipid arc (−25.2° vs. −35.7°, *p* = 0.004), mean angular extension of macrophages (−8.9° vs. −14.0°, *p* = 0.044) Colchicine significantly reduced Hs-CRP levels (geometric mean, 0.6 vs. 0.3, *p* = 0.046), IL-6 levels (geometric mean ratio, 0.8 vs. 0.5, *p* = 0.025), and myeloperoxidase levels (geometric mean ratio, 1.0 vs. 0.8, Pb = 0.047)
Vaidya et al., 2018 [70]	Prospective non-randomized observational study. Patients with recent ACS (<1 month), received either colchicine plus OMT or OMT alone	Primary: change in LAPV, a marker of plaque instability on CCTA and robust predictor of adverse cardiovascular events. Secondary: changes in other CCTA measures and in hs-CRP	0.5 mg daily	Colchicine = 40 Placebo = 40	12.6 months	Colchicine therapy significantly reduced LAPV (mean 15.9 mm^3^ [−40.9%] vs. 6.6 mm^3^ [−17.0%], *p* =0.008) and hs-CRP (mean 1.10 mg/L [−37.3%] vs. 0.38 mg/L [−14.6%], *p* < 0.001) vs. controls Atheroma volume (mean 42.3 mm^3^ vs. 26.4 mm^3^; *p* = 0.28) and low-density lipoprotein levels (mean 0.44 mmol/L vs. 0.49 mmol/l, *p* = 0.21) were similar between groups With multivariate linear regression, colchicine therapy remained significantly associated with greater reduction in LAPV (*p* = 0.039) and hs-CRP (*p* = 0.004) Significant linear association (*p* < 0.001) and strong positive correlation (*r* = 0.578) between change in LAPV and hs-CRP
Zuriaga et al., 2024 [71]	TET2-mutant clonal hematopoiesis was modeled in mice using bone marrow transplants in Ldlr−/− mice, treated with colchicine or placebo. In humans, data from two large biobanks were analyzed to assess if colchicine reduces the link between TET2 mutations and myocardial infarction		In mice: starting with 0.05 mg/kg/day for the first week, and transitioning to 0.1 mg/kg/day for the second week, and 0.2 mg/kg/day for the remaining 6 weeks	Humans Colchicine = 3849 Non colchicine users: 433,387	-	Mouse Model Colchicine reduced plaque size by ~27% in TET2-KO mice (*p* = 0.003). No effect on WT controls (*p* = 0.693) Colchicine reduced IL-1β production in TET2-KO macrophages more significantly than WT macrophages (48% vs. 16% decrease, *p* = 0.005) IL-6 reduction was similar in both genotypes (~70%) Human Studies Colchicine use attenuated the risk of MI (OR colchicine 0.76, 0.43–1.34 vs. OR no colchicine 1.23 (0.90–1.56, P_int_ = 0.04) Colchicine reduced MI (HR colchicine 0.30, 0.08–1.22 vs. HR no colchicine 1.08, 0.93–1.10, P_int_ = 0.05)

Abbreviations: CCS = Chronic Coronary Syndrome; ACS = Acute Coronary Syndrome; MI = Myocardial Infarction; PCI = Percutaneous Coronary Intervention; BMS = Bare-metal Stent; IVUS = Intravascular Ultrasound; CV = Cardiovascular; STEMI = ST-Elevation Myocardial Infarction; IS = Infarct Size; LV = Left Ventricle; MACEs = Major Adverse Cardiovascular Events; CK-MB = Creatine Kinase-Myocardial Brain; MRI = Magnetic Resonance Imaging; LGE = Late Gadolinium Enhancement; SCAI = Society for Cardiovascular Angiography and Interventions; NSTEMI = Non-ST-Elevation Myocardial Infarction; Hs-CRP = High-Sensitivity C-reactive Protein; OMT = Optimal Medical Therapy; IL = interleukin; LAPV = Low Attenuation Plaque Volume; CCTA = Coronary Computed Tomography Angiography; KO = Knock Out; WT = Wild Type.

## Data Availability

Data sharing is not applicable.

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
