# Peer review of "Combining Colchicine and Antiplatelet Therapy to Tackle Atherothrombosis: A Paradigm in Transition?"

_ijms, 2025, doi:10.3390/ijms26031136_

Round 1
Reviewer 1 Report
Comments and Suggestions for Authors
This is an interesting review discussing the medical uses of colchicine as "antiatherosclerotic" drug. The authors reviewed the current literature and provided background on the methodology for using this compound in the medical arena. I have only few minor issue:
1) the search strategy is not described. Please specifiy
2) Authors should expand the current guidelines (ACS 2023 and CCS 2024) since they are only cited. The level of recommendation remain low despite the growing evidence in the acute as well as in the chronic scenario. Please provide a deeper discussion on this section and how the use of colchicine could be improved in the current clinical scenario
Author Response
Please refer to the attached file.
RESPONSE TO REVIEWER #1
IJMS-3398852
Combining Colchicine and Antiplatelet therapy to Tackle Atherothrombosis:
A Paradigm in Transition?
This is an interesting review discussing the medical uses of colchicine as "antiatherosclerotic" drug. The authors reviewed the current literature and provided background on the methodology for using this compound in the medical arena. I have only few minor issue:
1) the search strategy is not described. Please specifiy
Author Reply: This is a narrative rather than a systematic review. By definition, narrative reviews do not require details on the research strategy. However, to comply with the Reviewer’s comment, we have added the following sentence at page 14: “We searched for in vitro and in vivo studies exploring the antiplatelet effects of colchicine in humans. Only studies in English were included. Table 1 summarizes the main characteristics and findings of included studies.”.
2) Authors should expand the current guidelines (ACS 2023 and CCS 2024) since they are only cited. The level of recommendation remain low despite the growing evidence in the acute as well as in the chronic scenario. Please provide a deeper discussion on this section and how the use of colchicine could be improved in the current clinical scenario.
Author Reply: We agree that a deeper discussion on guidelines recommendations is required. We also expanded on the current limitations in the use of colchicine in clinical practice. Accordingly, we have added at page 19 and 20 the following sentence: “Evidence from these RCTs led the 2023 ACS guidelines from the European Society of Cardiology (ESC) to recommend the use of low-dose colchicine (0.5 mg once a day) with a class IIb, level of evidence (LoE) A in ACS patients, particularly if other risk factors are insufficiently controlled or if recurrent cardiovascular disease events occur under optimal therapy (66). More recently, the 2024 CCS guidelines from the ESC recommended the use of low-dose colchicine with a class IIa, LoE A in CCS patients with atherosclerotic CAD to reduce MI, stroke, and need for revascularization (20). However, these guidelines were released before the publication of the largest CLEAR SYNERGY (OASIS 9) trial (58). CLEAR SYNERGY enrolled 7062 patients at 104 centers in 14 countries and found that after 5 years of follow-up, daily treatment with colchicine 0.5 mg daily in patients who suffered from an MI and underwent PCI, did not reduce MACE compared with placebo (58). Non univocal results also came from other clinical studies testing the effects of colchicine in other specific clinical settings. In brief, conflicting results on the efficacy of colchicine in reducing infarct size in patients with acute MI (67-69) and improving outcomes after PCI have been reported (57, 70, 71). On the other hand, colchicine has shown to promote atherosclerotic plaque stabilization (72, 73) and modulate clonal hematopoiesis (CH), a new emerging and independent cardiovascular risk factor (74). The conflicting clinical outcomes observed in large trials highlight, on one hand, the pressing need for more robust and conclusive clinical studies to reinforce guideline recommendations regarding its use. On the other hand, they emphasize the necessity for a deeper understanding of the pathophysiological mechanisms driving colchicine's effects, particularly in combination with antiplatelet therapy. One possible hypothesis is that only specific subsets of patients benefit from colchicine treatment. If this is the case, patient selection becomes crucial for implementing precision medicine approaches. From this perspective, pathophysiological studies aimed at enhancing our understanding of colchicine's mechanisms of action and its interactions with other medications, especially antiplatelet agents, are of the utmost importance. These investigations are essential not only to resolve uncertainties regarding efficacy but also to optimize the safe and targeted use of colchicine in clinical practice. Identifying the precise patient profiles most likely to benefit from colchicine could pave the way for personalized treatment strategies and improved outcomes. Finally, evidence comparing the safety and efficacy of a strategy combining colchicine with antiplatelet agents versus DAPT or DPI, alongside studies aimed at identifying the specific patient profiles most likely to benefit from dual targeting of inflammatory and platelet pathways rather than focusing solely on thrombotic pathways, could pave the way for personalized treatment approaches and better clinical outcomes.”.

Reviewer 2 Report
Comments and Suggestions for Authors
The article is a review of an interesting topic.
However, I believe this is not a very insightful review, and it does not contribute much in its current version.
It looks as if it was unfinished in places.
A review article - should systematize knowledge, not simplify it.
The layout of the work is not clear to me. I suggest introducing numbering so that individual chapters and subchapters are visible.
I suggest clearly distinguishing the goal.
Background
The authors wrote that atherosclerosis is a complex pathological process (but what does it mean?)
Is this not too much of an oversimplification?
Later in the text, the authors mentioned inflammation, but I think it is all mixed in this article and sounds chaotic.
Besides, what about oxidative stress - which has long been recognized as contributing to atherosclerosis?
The article lacks summary tables and non-obvious figures. There is one table in the article but no references to literature. Is this the researchers' original concept? Probably not - this should be corrected, i.e., supplemented.
As for the figure, the article contains one figure that looks more like a graphical abstract. This is good because a graphical abstract is needed, but there are still no figures explaining the mechanisms in this situation.
Although this is not a systematic review, I suggest that the authors add a few sentences about the selection of articles that they included.
I think the authors should write what databases they reviewed, what keywords they used, and what period of publication of the article they took into account. Did they include all the articles or only the results of studies on humans in English?
The authors cite their publications at least a dozen times (!). When I reviewed the download, the co-authorship of the cited works was approaching 20. This raises doubts in a review article, primarily since the authors did not provide the key according to which they selected the publications.
This raises my most considerable doubts.
Another serious omission is the lack of a Colchicine relationship, which prevents oxidative stress-induced endothelial cells (?).
The article does not contain clear conclusions. It does not set new trends or fully help understand the presented subject matter.
The article is written in one continuous sequence, and it seems it will not interest readers in its current version.
It should definitely be improved in terms of content and graphic design.
Author Response
Please refer to the attached file.
RESPONSE TO REVIEWER #2
IJMS-3398852
Combining Colchicine and Antiplatelet therapy to Tackle Atherothrombosis:
A Paradigm in Transition?
The article is a review of an interesting topic.
However, I believe this is not a very insightful review, and it does not contribute much in its current version. It looks as if it was unfinished in places. A review article - should systematize knowledge, not simplify it.
Author Reply: We thank the Reviewer for the valuable insights and the time invested in thoroughly reviewing our manuscript. We have now improved the paper as per the suggestions raised by you and the other Reviewers. We have also included two comprehensive tables to provide more details on the mentioned studies. We are confident that our paper is unique, as no other review article has specifically focused on the antiplatelet effects of colchicine, highlighting its originality within the field.
1) The layout of the work is not clear to me. I suggest introducing numbering so that individual chapters and subchapters are visible.
Author Reply: We have now numbered paragraphs as recommended.
2) I suggest clearly distinguishing the goal. Background: The authors wrote that atherosclerosis is a complex pathological process (but what does it mean?). Is this not too much of an oversimplification?
Author Reply: We have now modified the sentence as follows: “Atherosclerosis is a chronic and progressive disease of the arteries characterized by the accumulation of fibrofatty lesions, or plaques, within the arterial walls.”.
3) Later in the text, the authors mentioned inflammation, but I think it is all mixed in this article and sounds chaotic.
Author Reply: We agree that the subparagraph “inflammation and thrombosis” may generate confusion. Hence, it has been deleted and the whole paragraph has been shortened and revised as follows: “Recent evidence suggest that platelets, coagulation, and inflammatory processes are not three separate and independent entities, but rather part of a complex and interconnected system responsible for multiple physiological and pathophysiological functions, such as the immune response, anti-tumor response, thrombus formation, autoimmune diseases, and the progression and destabilization of atherosclerotic disease, including atherothrombosis (21, 22). A fundamental link between platelet, coagulation and inflammation is played by thrombin. In fact, according to the more recent “cell-based model of coagulation”, the extrinsic and intrinsic pathways of the coagulation cascade and platelets are sequentially involved and results in thrombin formation. Within this contest it is worth mentioning that activated platelets not only provide the best surface for the assembly of the coagulation factors, but they also express tissue factor (TF), the key activator of the blood coagulation cascade, becoming independent in triggering coagulation (23-25). Thrombin not only plays a central role in thrombus formation, but also drives a range of pro-inflammatory processes (21, 22, 26). These include the recruitment and activation of monocytes and neutrophils, induction of leukocyte adhesion molecules in endothelial cells, release of pro-inflammatory cytokines, and activation of the complement system (21, 22, 26). These processes synergistically contribute to both thrombosis and inflammation (13, 26). Several signaling pathways contribute to the crosstalk between thrombosis and inflammation, including protease-activated receptors (PARs), toll-like receptors (TLRs), complement, microparticles (MPs), and neutrophil extracellular traps (NETs) (9, 12, 27, 28). Among these, PAR-mediated signaling has been shown to play a central role (29). PARs belong to the G protein-coupled receptors (GPCRs) superfamily and have been identified in platelets, endothelial cells, leukocytes and vascular SMCs (12, 27). In humans, four members of the PAR family have been identified. PAR-1 is activated by FXa and thrombin (FIIa), PAR-2 by FXa and the FVIIa-TF complex, while PAR-3 and PAR-4 are activated by FIIa. Activation of PARs exerts significant effects on both platelets and inflammation. On the platelet side, PARs trigger the release of granules containing various chemokines, such as platelet factor 4, interleukin (IL)-1, IL-8, macrophage inflammatory protein, and Regulated upon Activation, Normal T cell Expressed and Secreted (RANTES). They also mobilize intracellular calcium and promote the expression of adhesion molecules like P-selectin and CD40 ligand, along with prothrombinase-binding phosphatidylserine on the platelet surface. This leads to platelet aggregation and pro-inflammatory effects, including the formation of platelet-leukocyte aggregates (12, 28). On the inflammatory side, PARs stimulate endothelial cells, monocytes, and fibroblasts to release pro-inflammatory molecules, including monocyte chemoattractant protein-1, tumor necrosis factor (TNF)α, and interleukins (IL-1β, IL-6, IL-8). They also activate endothelial cells, leading to the expression of adhesion molecules such as P- and E-selectins. Notably, there is a bidirectional relationship between coagulation, platelets, and inflammation, as pro-inflammatory interleukins (e.g., IL-1 and IL-6) can activate platelets and the coagulation cascade through multiple pathways (12, 27). Another connection between platelets and inflammation involves microparticles (MPs), which are primarily—but not exclusively—produced by activated platelets (30). Upon activation, platelet membrane asymmetry is lost, and the membrane detaches from the cytoskeleton, resulting in the formation of MPs. These MPs possess potent pro-inflammatory and pro-thrombotic properties. Inflammation influences thrombosis through several other mechanisms, such as endothelial injury, reduction of endogenous anticoagulants, inhibition of fibrinolytic activity, and enhanced pro-coagulant activity via TF-mediated coagulation activation (21, 22, 31). Healthy endothelium plays a crucial role in controlling hemostasis. Endothelial cells support natural anticoagulant processes by producing thrombomodulin, which activates protein C and deactivates thrombin. They also express heparan and dermatan sulfates, enhancing the activity of AT-III and heparin cofactors. Additionally, endothelial cells promote fibrinolysis by expressing TF pathway inhibitor (TFPI) and producing tPA and uPA. Furthermore, normal endothelium releases nitric oxide (NO) and prostacyclin (PGI2), leading to vasodilation. Inflammation disrupts this balance, increasing procoagulant and antifibrinolytic factors like von Willebrand factor (vWF), Thromboxane A2 (TxA2), plasminogen activator inhibitor type 1 (PAI-1), TF, and cell adhesion molecules, shifting the endothelium toward a proinflammatory and prothrombotic state. Procoagulant activity is normally regulated by three important anticoagulant pathways: antithrombin III (AT-III), the protein C system, and tissue TFPI. Evidence suggests that inflammation may negatively affect anticoagulant pathways such as AT-III and the protein C system. AT-III levels are reduced in chronic inflammation. AT-III protein is digested by elastase produced by neutrophils and consumed by thrombin. Moreover, AT-III activity is reduced due to the consumption of its co-factors (i.e., GAG) from endothelial surface. Inflammation also disrupts the protein C system. Protein C levels decrease due to reduced synthesis, increased consumption, and degradation by proteolytic enzymes like neutrophil elastase. Additionally, TNF-α and IL-1 downregulate thrombomodulin (TM), leading to reduced protein C activation. Inflammation also raises plasma levels of C4b-binding protein, which can cause a deficiency in free protein S, further promoting a procoagulant state. As far as it concerns the process of fibrinolysis, pro-inflammatory cytokines (i.e., TNF-α and IL-1) increase the release of fibrinolytic agents, such as tissue plasminogen activator (tPA) and urokinase-type plasminogen activator (uPA) from endothelial cells. This process, however, is counteracted by a sustained increase in PAI-1, resulting in an overall inadequate fibrin removal. TF is a transmembrane glycoprotein that plays a key role in initiating thrombin generation during inflammation, forming a critical link between inflammation and thrombosis. TF binds FVII leading to its activation into aFVII and catalyzes the conversion of FX into FXa directly or through the activation of FIX, ultimately generating thrombin. TF is primarily located in tissues not directly exposed to blood, such as the adventitial layer of blood vessels, having a primary role in the process of hemostasis (32). Inflammation increases the synthesis and expression of TF, mainly by endothelial cells, macrophages and platelets (31, 33). In atherosclerotic plaques, foam cells generate TF, which promotes thrombus formation upon plaque rupture (21, 31).”.
4) Besides, what about oxidative stress - which has long been recognized as contributing to atherosclerosis?
Author Reply: We acknowledge that oxidative stress plays a significant role in atherosclerosis. However, the referenced paragraph specifically addresses the interplay between coagulation, platelets, and inflammation in atherothrombosis. As such, delving into the pathophysiology of atherosclerosis formation lies beyond the scope of this manuscript.
5) The article lacks summary tables and non-obvious figures. There is one table in the article but no references to literature. Is this the researchers' original concept? Probably not - this should be corrected, i.e., supplemented.
Author Reply: We thank the Reviewer for this suggestion. To address this comment, we have removed table 1 and replaced it with two new comprehensive tables. Table 1 summarizes the main features and findings of in vivo and in vitro studies exploring the antiplatelet effects of colchicine (see below). Table 2 summarizes the main characteristics and findings of clinical studies exploring the effectiveness of colchicine in ASCVD patients (see below).
Table 1: Antiplatelet effects of colchicine.
|
Study Design |
Dosing |
Clinical setting |
Patients |
Main Findings |
|
|
In vitro-studies |
|
||||
|
Shah et al.
2016
|
Addition of colchicine to PRP (for 30 minutes) and whole blood (for 5 minutes) was followed by an evaluation of platelet activity and adhesion via LTA and flow cytometer |
0.015, 0.15, 1.5, 15, 150, 1500, 15000 μM
|
Healthy adults |
N= 10
|
Addition of colchicine to: · PRP significantly decreased platelet-platelet aggregation at >1.5 μM (maximal platelet aggregation 91% vs 84 %, P < 0.05)
· Whole blood decreased MPA (74.5% vs 51.1%, P= 0.04) and NPA (40.6% vs 26.4%, P=0.04) at 0.015 μM |
|
Brambilla et al.
2023
|
Whole blood was incubated with colchicine and then the expression of platelet-associated TF, P-selectin and GPIIbIIIa was measured by flow cytometry upon stimulation with ADP |
20 nM, 100 nm, 1 µM, 10 µM, 100 µM |
Healthy adults |
N=10 |
Colchicine reduced: · In a concentration-dependent manner: · GPIIbIIIa membrane expression (baseline vs 100 nM, P = 0.022; baseline vs 1 µM, P <0.0001; baseline vs 10 µM, P =0.0003; baseline vs 100 µM, P =0.0049) · ADP-induced TF externalization (baseline vs 1 µM, P =0.026; baseline vs 10 µM, P =0.001; baseline vs 100 µM, P =0.0002)
· At higher concentrations P-selectin expression (baseline vs 100 µM, P =0.022) |
|
Cirillo et al.
2020
|
PRP was pre-incubated with colchicine before being stimulated with ADP or TRAP. PRP not colchicine preincubated served as controls. The level of platelet aggregation was then evaluated by LTA at 30, 60 and 90 min |
10 μM |
Patients on DAPT with clopidogrel |
N=35 (28 clopidogrel responders and 7 clopidogrel non-responders) |
Colchicine: · Reduced TRAP-induced platelet aggregation in both clopidogrel responders (22 ± 7%; 19 ± 4%; 15 ± 1% [LTA-Platelet aggregation], P <0.05) and non-responders (20 ± 9%; 24 ± 8%; 22 ± 1%, P < 0.05) as compared to TRAP-stimulated platelets but not preincubated with colchicine
· Inhibited platelet aggregation in clopidogrel non-responders in which ADP still caused activation despite DAPT (22±12%, 19±11%, 21±8%, P <0.05) |
|
In vivo-studies |
|
||||
|
Shah et al.
2016
|
Administration of a 1.8 mg oral colchicine loading dose over one hour. Subsequent blood samples were drawn 2 and 24 hours after completion of the loading dose, platelet activity and adhesion were then assessed via LTA, flow cytometer and fluorescence microscope
|
1.8 mg over one hour |
Healthy adults |
N= 10
|
Colchicine: · Did not have significant effect on platelet aggregation in response to 1 μM ADP and 0.4 μM epinephrine (maximal platelet aggregation at baseline 84.4% vs 2 h 84.6% vs 24 h 87.5%; baseline vs 2 h, P =0.21; baseline vs 24 h, P =0.37)
· Significantly decreased MPA at 2 h but not at 24 h (baseline 27.8% vs 2 h 22% vs 24 h 35.6%; baseline vs 2 h, P =0.051; baseline vs 24 h, P =0.58)
· Significantly decreased NPA at 2 h but not at 24 h (baseline 18.9% vs 2 h 14.7% vs 24 h 21.8%; baseline vs 2 h, P =0.013; baseline vs 24 h, P =0.39)
· Reduced the platelet surface expression of PAC-1 at 2 h and 24 h (baseline 369.5 [mean fluorescence intensity] vs 2 h 332.5 vs 24 h 342.4; baseline vs 2 h, P =0.017; baseline vs 24 h, P =0.005) and of P-selectin at 2 hours but not at 24 h (baseline 350.6 vs 2 h 279.4 vs 24 h 311.8; baseline vs 2 h, P =0.03; baseline vs 24 h, P =0.44) |
|
Raju et al.
2011
|
Pilot randomized controlled trial comparing the effect of daily colchicine administration with placebo on hs-CRP levels and platelet function by turbidimetric platelet aggregometry |
1 mg/day for 30 days |
Patients with ACS or acute ischemic stroke |
N=80 |
Colchicine: · Did not significantly reduce absolute hs-CRP at 30 days (median 1.0 mg/l vs 1.5 mg/l, P = 0.22) · Did not affect platelet function assessed using platelet aggregation with ADP (5 μmol), arachidonic acid (0.5 mmol), collagen (1 μg/ml), collagen (5 μg/ml) and urine dehydrothromboxane B2, (P= 0.86, P = 0.64, P = 0.76, P = 0.20, respectively) |
|
Lee et al.
2023
|
Proof-of-concept pilot trial investigating the feasibility of ticagrelor or prasugrel P2Y12 inhibitor monotherapy combined with colchicine immediately after PCI in patients with ACS |
0.6 mg daily |
ACS patients treated with drug-eluting stents |
N=200 |
In ACS patients undergoing PCI, discontinuing aspirin therapy and administering low-dose colchicine on the day after PCI in addition to ticagrelor or prasugrel is associated with: · Low incidence of stent thrombosis (1.0% at 3 months); · Major bleeding is rare, (with a 3-month incidence of 0.5%;) · High platelet reactivity at discharge is low (0.5%); · Inflammatory levels were rapidly reduced within 1 month as shown by a significant decrease in hs-CRP levels (from 6.1 mg/L at 24 hours after PCI to 0.6 mg/L at 1 month, P < 0.001) |
Abbreviations: PRP=Platelet Rich Plasma; LTA=Light Transmission Aggregometry; MPA=Monocyte Platelet Aggregation; NPA=Neutrophil Platelet Aggregation; TF=Tissue Factor; GPIIbIIIa= Glicoprotein IIbIIIa; ADP= Adenosine Diphosphate; TRAP=Thrombin Receptor Activating Peptide; DAPT=Dual Antiplatelet Therapy; hs-CRP= High Sensivity C-reactive Protein; ACS= Acute Coronary Syndrome; PCI=Percoutaneous Coronary Intervention.
Table 2: Main characteristics and findings of clinical studies exploring the effectiveness of colchicine in ASCVD patients
|
Study Design |
Outcomes |
Dosing |
Sample size |
Follow-up |
Main Findings |
|
|
Nidorf et al.
2013
|
Randomized, observer-blinded trial. CCS patients were assigned to colchicine or no colchicine |
Primary composite: ACS, out of hospital cardiac arrest, or non- cardioembolic stroke
Secondary: individual components of the primary outcome and the components of ACS unrelated to stent disease |
0.5 mg/day |
Colchicine= 282
Controls= 250 |
3 years |
· Colchicine reduced the primary outcome (5.3% vs 16.0%, HR 0.33, 0.18-0.59, P< 0.001) compared with placebo · Colchicine reduced the incidence of ACS (4.6% vs 13.6%, HR 0.33, 0.18-0.63, P <0.001) and non- cardioembolic stroke (0.35% vs 1.6%, HR 0.23, 0.03-2.03, P =0.011) compared with placebo · In a pre-specified secondary on-treatment analysis that excluded 32 patients assigned to colchicine who withdrew within 30 days due to intestinal intolerance and a further 7 patients (2%) who did not start treatment, the primary outcome occurred less frequently with colchicine compared to placebo (4.5% vs 16.0%, HR 0.29, 0.15-0.56, P< 0.001) |
|
Nidorf et al.
2020
|
Randomized, controlled, double-blind trial. CCS participants were assigned to receive either colchicine or placebo |
Primary composite: cardiovascular death, spontaneous (nonprocedural) MI, ischemic stroke, or ischemia-driven coronary revascularization. Secondary composite: cardiovascular death, spontaneous MI, or ischemic stroke |
0.5 mg/day or no colchicine |
Colchicine= 2762
Controls= 2760 |
28.6 months |
· Colchicine reduced the primary outcome compared with placebo (6.8% vs 9.6%, HR 0.69, 0.57-0.83, P <0.001) · Colchicine reduced the key secondary endpoint (4.2% vs 5.7%, HR 0.72, 0.57-0.92, P =0.007) compared with placebo |
|
Tardif et al.
2019
|
Randomized, double-blind trial involving patients recruited within 30 days after a MI. The patients were randomly assigned to receive either low-dose colchicine or placebo |
Primary composite: death from cardiovascular causes, resuscitated cardiac arrest, MI, stroke, or urgent hospitalization for angina leading to coronary revascularization. Secondary: consisted of the components of the primary end point; a composite of death from cardiovascular causes, resuscitated cardiac arrest, myocardial infarction, or stroke; and total mortality in time-to-event analyses |
0.5 mg/day or placebo |
Colchicine=2366 Placebo= 2379 |
22.6 months |
· Colchicine significantly reduced the primary endpoint (5.5% vs 7.1%, HR 0.77, 0.61-0.96, P =0.02) compared to placebo · Colchicine reduced stroke (HR 0.26, 0.10-0.70), and urgent hospitalization for angina leading to coronary revascularization (HR 0.50, 0.31-0.81), compared to placebo · No difference in the secondary composite endpoints between groups (4.7% vs 5.5%) · No difference in total mortality between groups (1.8% vs 1.8%) · No difference in the incidence of diarrhea between groups (9.7% vs 8.9%, P =0.35) · Increased risk of pneumonia with colchicine compared to placebo (0.9% vs 0.4%, P = 0.03) |
|
Tong et al.
2020
|
Multicenter, randomized, double-blind, placebo-controlled trial. Patients who presented with ACS and had evidence of coronary artery disease on coronary angiography managed with either PCI or medical therapy were assigned to receive either colchicine or placebo
|
Primary composite: all-cause mortality, ACS, ischemia-driven (unplanned) urgent revascularization, and non-cardioembolic ischemic stroke in a time to event analysis
|
0.5 mg twice daily for the first month, then 0.5 mg daily for 11 months or placebo |
Colchicine= 396 Placebo= 399 |
12 months |
· No difference in the primary outcome between groups (24 vs 38 events, P =0.09) · Increase in total death (8 vs 1, P =0.017) and non-CV death (5 vs 0, P =0.024) with colchicine · No difference in adverse effects between groups (23.0% vs 24.3%). The majority of them were gastrointestinal symptoms (23.0% vs 20.8%) |
|
Deftereos et al.
2013
|
Double-blind, prospective, placebo-controlled study. Diabetic patients with contraindication to a drug-eluting stent, undergoing PCI with a BMS, were randomized to colchicine or placebo. Angiography and intravascular ultrasound were performed 6 months after the index PCI |
Primary: Angiographic and IVUS restenosis
Secondary: angiographic and IVUS parameters of lumen loss and in-stent neointimal hyperplasia |
0.5 mg twice daily or placebo |
Colchicine= 100 Placebo=110 |
6 months |
· Colchicine significantly reduced angiographic restenosis (16% vs 33%, P =0.007) and IVUS restenosis (24% vs 43%, P =0.006) · Colchicine significantly reduced minimum lumen diameter (2.8 mm (2.2–3.1) vs 2.3 mm (1.3-2.9), P <0.01) · Gastrointestinal symptoms (diarrhea and nausea): were the most common adverse events in the colchicine group (16% vs 7%, P =0.058) |
|
Opstal et al.
2023
|
Randomized, parallel, double-blind trial that evaluated the effect of adding colchicine or placebo in patients with chronic coronary disease |
Cause-specific mortality data were analysed, stratified by treatment status |
0.5 mg once daily or placebo |
Colchicine=2762 Placebo=2760 |
29 months |
· No difference in CV death between groups (0.7% vs 0.9%)
· No difference in non-CV death between groups (1.9% vs 1.3%)
|
|
Jolly et al.
2024
|
Multicenter trial with a 2-by-2 factorial design. It was randomly assigned patients who had myocardial infarction to receive either colchicine or placebo and either spironolactone or placebo. |
Primary composite: death from cardiovascular causes, recurrent MI, stroke, or unplanned ischemia-driven coronary revascularization |
For the first 90 days: patients weighing >70 kg 0.5 mg x 2/day, if <. 70 kg 0.5 mg /day. After 90 days 0.5 mg /day for all patients |
Colchicine= 3528 Placebo= 3534 |
2.98 years |
· No difference in the primary-outcome between groups (9.1% vs 9.3%, P =0.93)
· Colchicine increased the incidence of diarrhea (10.2% vs 6.6%, P <0.001)
|
|
Mewton et al.
2021
|
Double-blind multicenter trial. Patients admitted for a first episode of STEMI referred for PCI were randomized to receive colchicine or placebo from admission to day 5. Patients underwent a cardiac magnetic resonance at 5 days and at 30 days |
Primary: reduction of IS at 5 days.
Secondary: LV end-diastolic volume change at 3 months and IS at 3 months |
2 mg loading dose followed by 0.5 mg twice a day for 5 days |
Colchicine= 101 Placebo= 91 |
3 months |
· At 5 days IS did not differ between the colchicine and placebo groups (26 g vs 28.4 g of LV mass, P = 0.87) · At 3 months follow-up, there was no significant difference in LV remodeling (colchicine +2.4% vs –1.1% change in LV end-diastolic volume, P =0.49) · Infarct size at 3 months was also not significantly different between the colchicine and placebo groups (17g vs 18 g, P =0.92) · The incidence of gastrointestinal adverse events during the treatment period was greater with colchicine than with placebo (34% vs 11%, P =0.0002) |
|
Bouleti et al.
2023
|
Follow-up analysis of the COVERT-MI study on prespecified secondary clinical endpoints |
Primary composite: all-cause death, ACS, heart failure events, ischaemic strokes, sustained ventricular arrhythmias and acute kidney injury |
2 mg loading dose followed by 0.5 mg two times a day for 5 days or placebo |
Colchicine= 101 Placebo=91 |
1 year |
· No significant difference regarding the number of MACEs between groups · No differences in the occurrence of ischemic strokes: colchicine 3% vs placebo 2.2% (P =0.99) |
|
Deftereos et al.
2015
|
Prospective, double-blinded, placebo-controlled study. Patients presenting with STEMI ≤12 hours from pain onset (treated with PCI) were randomly assigned to colchicine or placebo for 5 days. A subset of patients underwent cardiac MRI 6 to 9 days after the index STEMI (MRI subgroup) |
Primary: area under the curve of CK-MB fraction concentration over 72 hours after admission
Secondary: Maximal high-sensitivity troponin T measure during the same time period.
In MRI subgroup absolute MI volume, determined by LGE, was the primary outcome measure |
Loading dose of 2 mg (1.5 mg initially followed by 0.5 mg 1 hour later) and continuing with 0.5 mg twice daily, or placebo, for 5 days |
Colchicine= 77 Placebo=74 MRI subgroup= 60
|
5 days, until 9 days for MRI subgroup |
· The area under the creatine kinase-myocardial brain fraction curve was for colchicine group 3144 ng·h–1·mL–1 vs placebo group: 6184 ng·h–1·mL–1, P <0.001 · Median maximum high-sensitivity troponin T was 19763 pg/mL and 45 550 pg/mL in the colchicine and control group, respectively (P =0.001) · Indexed MRI-late gadolinium enhancement–defined infarct size was 18.3 mL/1.73 m2 in the colchicine vs 23.2 mL/1.73 m2 in control group (P =0.019) · The relative infarct size (as a proportion to left ventricular myocardial volume) was 13.0 % in the colchicine and 19.8 %, in the control group (P =0.034) |
|
Shah et al.
2020
|
Randomized, double-blind, placebo-controlled trial. Subjects referred for possible PCI were randomized to acute preprocedural oral administration of colchicine or placebo |
Primary: PCI-related myocardial injury according to the Universal Definition
Secondary: Occurrence of 30-day MACEs (earliest occurrence of death from any cause, nonfatal MI, or target vessel revascularization) PCI-related MI as defined by the SCAI (76) |
1.2 mg 1 to 2 hours before coronary angiography, followed by colchicine 0.6 mg 1 hour later or immediately preprocedure |
Colchicine= 366 Placebo=348
Colchicine+PCI= 206 Placebo+PCI= 194 |
30 days |
· Primary outcome did not differ between colchicine and placebo groups (57.3% vs 64.2%, P =0.19) · Secondary composite outcome: colchicine 11.7% vs placebo 12.9% (P =0.82) · PCI-related MI defined by the SCAI (colchicine 2.9% vs placebo 4.7%, P =0.49) |
|
Cole et al.
2021
|
Randomized pilot trial. Patients undergoing PCI for stable angina or NSTEMI were randomized to oral colchicine or placebo, 6 to 24 hours pre-procedure |
Primary: periprocedural myocardial infarction |
1 mg followed by 0.5 mg 1 hour later |
Colchicine= 36 Placebo= 39
|
24 h |
· No patients developed periprocedural myocardial infarction in either group · Colchicine significantly reduced major periprocedural myocardial -injury: colchicine 31% vs placebo 54%, P =0.04 · Colchicine significantly reduced minor periprocedural myocardial -injury: colchicine 58% vs placebo 85%, P =0.01 |
|
Yu et al.
2024
|
Prospective, single-center, randomized, double-blind clinical trial. Patients with ACS with lipid-rich plaque detected by optical coherence tomography were included. The subjects were randomly assigned to receive either colchicine or placebo |
Primary: Change in the minimal fibrous cap thickness from baseline to the 12-month follow-up
|
0.5 mg once daily |
Colchicine= 52 Placebo=52
|
12 months |
· Colchicine increased the minimal fibrous cap thickness (51.9 μm vs 87.2 μm, P =0.006), and reduced average lipid arc (–25.2° vs –35.7°, P =0.004), mean angular extension of macrophages (–8.9° vs –14.0°, P =0.044) · Colchicine significantly reduced Hs-CRP levels (geometric mean, 0.6 vs 0.3, P =0.046), interleukin-6 levels (geometric mean ratio, 0.8 vs 0.5, P =0.025), and myeloperoxidase levels (geometric mean ratio, 1.0 vs 0.8, Pb =0.047)
|
|
Vaidya et al.
2018
|
Prospective non-randomized observational study. Patients with recent ACS (<1 month), received either colchicine plus OMT or OMT alone |
Primary: change in LAPV a marker of plaque instability on CCTA and robust predictor of adverse cardiovascular events.
Secondary: changes in other CCTA measures and in hs-CRP |
0.5 mg/day plus OMT or OMT alone |
Colchicine= 40 Placebo=40
|
12.6 months |
· Colchicine therapy significantly reduced LAPV (mean 15.9 mm3 [−40.9%] vs 6.6 mm3 [−17.0%], P =0.008) and hs-CRP (mean 1.10 mg/l [−37.3%] vs 0.38 mg/l [−14.6%], P <0.001) vs controls · Atheroma volume (mean 42.3 mm3 vs 26.4 mm3; P =0.28) and low-density lipoprotein levels (mean 0.44 mmol/l vs 0.49 mmol/l, P =0.21) were similar between groups · With multivariate linear regression, colchicine therapy remained significantly associated with greater reduction in LAPV (P =0.039) and hs-CRP (P =0.004) · Significant linear association (P <0.001) and strong positive correlation (r =0.578) between change in LAPV and hs-CRP
|
|
Zuriaga et al.
2024
|
TET2-mutant clonal haematopoiesis was modeled in mice using bone marrow transplants in Ldlr−/− mice, treated with colchicine or placebo. In humans, data from two large biobanks were analyzed to assess if colchicine reduces the link between TET2 mutations and myocardial infarction |
|
In mice: starting with 0.05 mg/kg/day for the first week, and transitioning to 0.1 mg/kg/day for the second week, and 0.2 mg/kg/day for the remaining 6 weeks |
Humans Colchicine=3849 Non cholchicine users: 433387
|
- |
Mouse Model · Colchicine reduced plaque size by ~27% in TET2-KO mice, P =0.003 No effect on WT controls, P =0.693 · Colchicine reduced IL-1β production in TET2-KO macrophages more significantly than WT macrophages (48% vs 16% decrease, P = 0.005) · IL-6 reduction was similar in both genotypes (~70%)
Human Studies · Colchicine use attenuated the risk of MI (OR colchicine 0.76, 0.43–1.34 vs OR no colchicine 1.23 (0.90-1.56, Pint= 0.04) · Colchicine reduced MI (HR colchicine 0.30, 0.08–1.22 vs HR no colchicine 1.08, 0.93-1.10, Pint= 0.05)
|
Abbreviations: CCS= Chronic Cornary Syndrome; ACS= Acute Coronary Syndrome; MI= Myocardial Infarction; PCI=Percoutaneous Coronary Intervention; BMS=Bare-metal Stent; IVUS=Intravascular Ultrasound; CV= Cardiovascular; STEMI =ST-Elevation Myocardial Infarction; IS=Infarct Size; LV=Left Ventricle; MACEs= Major Adverse Cardiovascular Events; CK-MB=Creatine Kinase-Myocardial Brain; MRI=Magnetic Resonance Imaging; LGE= Late Gadolinium Enhancement; SCAI=Society for Cardiovascular Angiography and Interventions; NSTEMI= Non ST-Elevation Myocardial Infarction; Hs-CRP= High Sensitivity C-reactive Protein; OMT=Optimal Medical Therapy; LAPV =Low Attenuation Plaque Volume; CCTA= Coronary Computed Tomography Angiography; KO=Knock Out; WT=Wild Type.
6) As for the figure, the article contains one figure that looks more like a graphical abstract. This is good because a graphical abstract is needed, but there are still no figures explaining the mechanisms in this situation.
Author Reply: We agree that our figure serves as a graphical abstract, effectively capturing readers' interest while providing a clear and concise overview of the paper's message. The aim of this study is to summarize the available clinical and pre-clinical evidence on the combined use of colchicine with antiplatelet therapy to address atherothrombosis. Moreover, we contextualized this strategy within the framework of other currently recommended approaches, such as dual antiplatelet therapy and dual-pathway inhibition. We believe the figure appropriately supports this objective, while also referencing the main pathophysiological processes underlying atherosclerotic formation and destabilization. In order to better define the aim of our article, we have added the following sentence in the abstract: “However, many aspects of colchicine’s mechanism of action, including its antiplatelet effects and how it synergizes with antiplatelet therapy remain unclear. In this review, we summarize the available clinical and pre-clinical evidence on the antiplatelet effects of colchicine and its synergistic interactions with antiplatelet therapy, highlighting their potential role in addressing atherothrombosis.”; as well as in the introduction paragraph at page 5 and 6: “However, colchicine’s mechanism of action remains incompletely understood. Emerging evidence suggests that colchicine may also exhibit antiplatelet properties, indicating that its cardiovascular benefits may at least partially stem from its ability to inhibit platelet activity. These mechanisms may be particularly relevant in clinical practice, especially considering that colchicine is frequently added to antiplatelet therapy in patients with ASCVD. This combination has the potential to influence not only thrombotic but also bleeding events. In this review, we summarize the available clinical and pre-clinical evidence on the antiplatelet effects of colchicine and its synergistic interactions with antiplatelet therapy, highlighting their potential role in addressing atherothrombosis.”.
7) Although this is not a systematic review, I suggest that the authors add a few sentences about the selection of articles that they included.
Author Reply: We agree with the Reviewer that narrative reviews do not require details on the research strategy. However, to comply with the Reviewer’s comment, we have added the following sentence at page 14: “We searched for in vitro and in vivo studies exploring the antiplatelet effects of colchicine in humans. Only studies in English were included. Table 1 summarizes the main characteristics and findings of included studies.”.
8) I think the authors should write what databases they reviewed, what keywords they used, and what period of publication of the article they took into account. Did they include all the articles or only the results of studies on humans in English?
Author Reply: Please refer to the previous point.
9) The authors cite their publications at least a dozen times (!). When I reviewed the download, the co-authorship of the cited works was approaching 20. This raises doubts in a review article, primarily since the authors did not provide the key according to which they selected the publications.
Author Reply: As noted, this is not a systematic review; therefore, it does not require a database or a defined research strategy. Nonetheless, we believe that all relevant studies on the antiplatelet effects of colchicine as well as all the main clinical studies on the effectiveness of colchicine in ASCVD patients have been included. If we have overlooked any pertinent studies, we welcome specific suggestions. Similarly, if the Reviewer considers any of the cited references inappropriate, we kindly request specific feedback on why a particular reference may not be suitable. The co-authors of this manuscript include researchers who have significantly contributed to this field, making it reasonable that some of the cited works are authored by them.
10) This raises my most considerable doubts. Another serious omission is the lack of a Colchicine relationship, which prevents oxidative stress-induced endothelial cells (?).
Author Reply: As stated above, we agree that oxidative stress plays a significant role in atherosclerosis and that there is some evidence showing that colchicine prevents oxidative stress-induced endothelial cell (10.1186/s12950-023-00366-7). However, our review focuses on the antiplatelet effects of colchicine and its potential role in tackling atherothrombosis. Therefore, we believe that expanding on the possible role of colchicine in preventing oxidative stress-induced endothelial cell goes beyond the scope of this manuscript.
11) The article does not contain clear conclusions. It does not set new trends or fully help understand the presented subject matter.
Author Reply: Thank you for this suggestion. We have now modified the paragraph “Current evidence and future perspectives” in order to provide more comprehensive conclusions, including a deeper discussion on guideline recommendations and on current limitations in the use of colchicine in clinical practice. Accordingly, the paragraph now reads as follows: “Colchicine has shown a number of in vitro and in vivo effects that support the rationale for its use in the treatment and prevention of atherothrombosis in patients with atherosclerotic cardiovascular disease, particularly those with coronary artery disease. The association of colchicine and a single or dual antiplatelet therapy, particularly a P2Y12 inhibitor, may represent a new frontier to tackle atherothrombosis with a more favorable safety profile compared other strategies focusing on platelets and coagulation pathways, such as DAPT or DPI strategies (Central figure). Further research is needed to elucidate the pathophysiological mechanisms underlying the combined use of colchicine and antiplatelet agents. This includes exploring the interactions between colchicine's anti-inflammatory properties and the antithrombotic effects of antiplatelet therapies, as well as their collective impact on platelet function, vascular inflammation, and thrombotic pathways. Additionally, robust clinical studies are essential to provide comprehensive evidence supporting the safety, efficacy, and optimal therapeutic regimens for this combination strategy. Indeed, thus far clinical studies evaluating the effects of colchicine in patients with CAD have yielded mixed results (Table 2). Five pivotal randomized controlled trials have investigated whether the addition of colchicine on top of optimal medical therapy could improve the outcomes for primary and secondary prevention of patients affected by ASCVD (54-58). Overall, these trials found that colchicine reduced MACE by 25%, myocardial infarction by 22%, stroke by 46%, and coronary revascularization by 23% in patients with both ACS and CCS (64). Although some concern on the possible higher rate of non-CV death in the colchicine vs placebo group was raised, a subsequent sub-study of the Low-Dose Colchicine 2 (LoDoCo2) trial found no significant association with any specific causes of death (65, 66). Evidence from these RCTs led the 2023 ACS guidelines from the European Society of Cardiology (ESC) to recommend the use of low-dose colchicine (0.5 mg once a day) with a class IIb, level of evidence (LoE) A in ACS patients, particularly if other risk factors are insufficiently controlled or if recurrent cardiovascular disease events occur under optimal therapy (67). More recently, the 2024 CCS guidelines from the ESC recommended the use of low-dose colchicine with a class IIa, LoE A in CCS patients with atherosclerotic CAD to reduce MI, stroke, and need for revascularization (20). However, these guidelines were released before the publication of the largest CLEAR SYNERGY (OASIS 9) trial (59). CLEAR SYNERGY enrolled 7062 patients at 104 centers in 14 countries and found that after 5 years of follow-up, daily treatment with colchicine 0.5 mg daily in patients who suffered from an MI and underwent PCI, did not reduce MACE compared with placebo (59). Non univocal results also came from other clinical studies testing the effects of colchicine in other specific clinical settings. In brief, conflicting results on the efficacy of colchicine in reducing infarct size in patients with acute MI (68-70) and improving outcomes after PCI have been reported (58, 71, 72). On the other hand, colchicine has shown to promote atherosclerotic plaque stabilization (73, 74) and modulate clonal hematopoiesis (CH), a new emerging and independent cardiovascular risk factor (75). The conflicting clinical outcomes observed in large trials highlight, on one hand, the pressing need for more robust and conclusive clinical studies to reinforce guideline recommendations regarding its use. On the other hand, they emphasize the necessity for a deeper understanding of the pathophysiological mechanisms driving colchicine's effects, particularly in combination with antiplatelet therapy. One possible hypothesis is that only specific subsets of patients benefit from colchicine treatment. If this is the case, patient selection becomes crucial for implementing precision medicine approaches. From this perspective, pathophysiological studies aimed at enhancing our understanding of colchicine's mechanisms of action and its interactions with other medications, especially antiplatelet agents, are of the utmost importance. These investigations are essential not only to resolve uncertainties regarding efficacy but also to optimize the safe and targeted use of colchicine in clinical practice. Identifying the precise patient profiles most likely to benefit from colchicine could pave the way for personalized treatment strategies and improved outcomes. Finally, evidence comparing the safety and efficacy of a strategy combining colchicine with antiplatelet agents versus DAPT or DPI, alongside studies aimed at identifying the specific patient profiles most likely to benefit from dual targeting of inflammatory and platelet pathways rather than focusing solely on thrombotic pathways, could pave the way for personalized treatment approaches and better clinical outcomes.”.
12) The article is written in one continuous sequence, and it seems it will not interest readers in its current version. It should definitely be improved in terms of content and graphic design.
Author Reply: we hope that the current version, that has been extensively implemented according to the comments by the Editors and Reviewers, now satisfies the Reviewer.

Reviewer 3 Report
Comments and Suggestions for Authors
1. Is the article aiming to evaluate dual-pathway inhibition, inflammation-targeting strategies, or both? A more precise focus will improve the flow.
2. Incorporate specific data points, such as statistics or study results, to substantiate claims about the efficacy and safety of different therapeutic strategies. This will add rigor to the article.
3. Address the bleeding risk associated with dual antiplatelet therapy and dual-pathway inhibition in more detail. Consider stratifying patients based on bleeding risk to contextualize the strategies better.
4. Provide more practical insights for clinicians. For instance, offer guidance on how to select between dual antiplatelet therapy, dual-pathway inhibition, and colchicine for specific patient populations.
Author Response
Please refer to the attahced file.
RESPONSE TO REVIEWER #3
IJMS-3398852
Combining Colchicine and Antiplatelet therapy to Tackle Atherothrombosis:
A Paradigm in Transition?
- Is the article aiming to evaluate dual-pathway inhibition, inflammation-targeting strategies, or both? A more precise focus will improve the flow.
Author Reply: We thank the Reviewer for this comment. The aim of this study is to summarize the available clinical and pre-clinical evidence on the combined use of colchicine with antiplatelet therapy to address atherothrombosis. Moreover, we contextualized this strategy within the framework of other currently recommended approaches, such as dual antiplatelet therapy and dual-pathway inhibition. However, we agree that the aim of our manuscript was not clear in the original form. In order to better define the aim of our article, we have added the following sentence in the abstract: “However, many aspects of colchicine’s mechanism of action, including its antiplatelet effects and how it synergizes with antiplatelet therapy remain unclear. In this review, we summarize the available clinical and pre-clinical evidence on the antiplatelet effects of colchicine and its synergistic interactions with antiplatelet therapy, highlighting their potential role in addressing atherothrombosis.”; as well as in the introduction paragraph at page 5 and 6: “However, colchicine’s mechanism of action remains incompletely understood. Emerging evidence suggests that colchicine may also exhibit antiplatelet properties, indicating that its cardiovascular benefits may at least partially stem from its ability to inhibit platelet activity. These mechanisms may be particularly relevant in clinical practice, especially considering that colchicine is frequently added to antiplatelet therapy in patients with ASCVD. This combination has the potential to influence not only thrombotic but also bleeding events. In this review, we summarize the available clinical and pre-clinical evidence on the antiplatelet effects of colchicine and its synergistic interactions with antiplatelet therapy, highlighting their potential role in addressing atherothrombosis.”.
- Incorporate specific data points, such as statistics or study results, to substantiate claims about the efficacy and safety of different therapeutic strategies. This will add rigor to the article.
Author Reply: We agree that reporting more specifics and results about the clinical and pre-clinical studies on the topic will add rigor to the study and improve its impact. Accordingly, we have included two detailed tables, one (Table 1) summarizing the main features and findings of in vivo and in vitro studies exploring the antiplatelet effects of colchicine (see below) and the other (Table 2) summarizing the main characteristics and findings of clinical studies exploring the effectiveness of colchicine in ASCVD patients (see below). We have decided to include these details in the tables given the fact that this is a narrative with limited word count.
Table 1: Antiplatelet effects of colchicine.
|
Study Design |
Dosing |
Clinical setting |
Patients |
Main Findings |
|
|
In vitro-studies |
|
||||
|
Shah et al.
2016
|
Addition of colchicine to PRP (for 30 minutes) and whole blood (for 5 minutes) was followed by an evaluation of platelet activity and adhesion via LTA and flow cytometer |
0.015, 0.15, 1.5, 15, 150, 1500, 15000 μM
|
Healthy adults |
N= 10
|
Addition of colchicine to: · PRP significantly decreased platelet-platelet aggregation at >1.5 μM (maximal platelet aggregation 91% vs 84 %, P < 0.05)
· Whole blood decreased MPA (74.5% vs 51.1%, P= 0.04) and NPA (40.6% vs 26.4%, P=0.04) at 0.015 μM |
|
Brambilla et al.
2023
|
Whole blood was incubated with colchicine and platelet-associated TF, P-selectin and GPIIbIIIa, then their expression was measured by flow cytometry upon stimulation with ADP |
20 nM, 100 nm, 1 µM, 10 µM, 100 µM |
Healthy adults |
N=10 |
Colchicine reduced: · In a concentration-dependent manner: · GPIIbIIIa membrane expression (baseline vs 100 nM, P = 0.022; baseline vs 1 µM, P <0.0001; baseline vs 10 µM, P =0.0003; baseline vs 100 µM, P =0.0049) · ADP-induced TF externalization (baseline vs 1 µM, P =0.026; baseline vs 10 µM, P =0.001; baseline vs 100 µM, P =0.0002)
· At higher concentrations P-selectin expression (baseline vs 100 µM, P =0.022) |
|
Cirillo et al.
2020
|
PRP was pre-incubated with colchicine before being stimulated with ADP or TRAP. PRP not colchicine preincubated served as controls. The level of platelet aggregation was then evaluated by LTA at 30, 60 and 90 min |
10 μM |
Patients on DAPT with clopidogrel |
N=35 (28 clopidogrel responders and 7 clopidogrel non-responders) |
Colchicine: · Reduced TRAP-induced platelet aggregation in both clopidogrel responders (22 ± 7%; 19 ± 4%; 15 ± 1% [LTA-Platelet aggregation], P <0.05) and non-responders (20 ± 9%; 24 ± 8%; 22 ± 1%, P < 0.05) as compared to TRAP-stimulated platelets but not preincubated with colchicine
· Inhibited platelet aggregation in clopidogrel non-responders in which ADP still caused activation despite DAPT (22±12%, 19±11%, 21±8%, P <0.05) |
|
In vivo-studies |
|
||||
|
Shah et al.
2016
|
Administration of a 1.8 mg oral colchicine loading dose over one hour. Subsequent blood samples were drawn 2 and 24 hours after completion of the loading dose, platelet activity and adhesion were then assessed via LTA, flow cytometer and fluorescence microscope
|
1.8 mg over one hour |
Healthy adults |
N= 10
|
Colchicine: · Did not have significant effect on platelet aggregation in response to 1 μM ADP and 0.4 μM epinephrine (maximal platelet aggregation at baseline 84.4% vs 2 h 84.6% vs 24 h 87.5%; baseline vs 2 h, P =0.21; baseline vs 24 h, P =0.37)
· Significantly decreased MPA at 2 h but not at 24 h (baseline 27.8% vs 2 h 22% vs 24 h 35.6%; baseline vs 2 h, P =0.051; baseline vs 24 h, P =0.58)
· Significantly decreased NPA at 2 h but not at 24 h (baseline 18.9% vs 2 h 14.7% vs 24 h 21.8%; baseline vs 2 h, P =0.013; baseline vs 24 h, P =0.39)
· Reduced the platelet surface expression of PAC-1 at 2 h and 24 h (baseline 369.5 [mean fluorescence intensity] vs 2 h 332.5 vs 24 h 342.4; baseline vs 2 h, P =0.017; baseline vs 24 h, P =0.005) and of P-selectin at 2 hours but not at 24 h (baseline 350.6 vs 2 h 279.4 vs 24 h 311.8; baseline vs 2 h, P =0.03; baseline vs 24 h, P =0.44) |
|
Raju et al.
2011
|
Pilot randomized controlled trial comparing the effect of daily colchicine administration with placebo on hs-CRP levels and platelet function by turbidimetric platelet aggregometry |
1 mg/day for 30 days |
Patients with ACS or acute ischemic stroke |
N=80 |
Colchicine: · Did not significantly reduce absolute hs-CRP at 30 days (median 1.0 mg/l vs 1.5 mg/l, P = 0.22) · Did not affect platelet function assessed using platelet aggregation with ADP (5 μmol), arachidonic acid (0.5 mmol), collagen (1 μg/ml), collagen (5 μg/ml) and urine dehydrothromboxane B2, (P= 0.86, P = 0.64, P = 0.76, P = 0.20, respectively) |
|
Lee et al.
2023
|
Proof-of-concept pilot trial investigating the feasibility of ticagrelor or prasugrel P2Y12 inhibitor monotherapy combined with colchicine immediately after PCI in patients with ACS |
0.6 mg daily |
ACS patients treated with drug-eluting stents |
N=200 |
In ACS patients undergoing PCI, discontinuing aspirin therapy and administering low-dose colchicine on the day after PCI in addition to ticagrelor or prasugrel is associated with: · Low incidence of stent thrombosis (1.0% at 3 months); · Major bleeding is rare, (with a 3-month incidence of 0.5%;) · High platelet reactivity at discharge is low (0.5%); · Inflammatory levels were rapidly reduced within 1 month as shown by a significant decrease in hs-CRP levels (from 6.1 mg/L at 24 hours after PCI to 0.6 mg/L at 1 month, P < 0.001) |
Abbreviations: PRP=Platelet Rich Plasma; LTA=Light Transmission Aggregometry; MPA=Monocyte Platelet Aggregation; NPA=Neutrophil Platelet Aggregation; TF=Tissue Factor; GPIIbIIIa= Glicoprotein IIbIIIa; ADP= Adenosine Diphosphate; TRAP=Thrombin Receptor Activating Peptide; DAPT=Dual Antiplatelet Therapy; hs-CRP= High Sensivity C-reactive Protein; ACS= Acute Coronary Syndrome; PCI=Percoutaneous Coronary Intervention.
Table 2: Main characteristics and findings of clinical studies exploring the effectiveness of colchicine in ASCVD patients
|
Study Design |
Outcomes |
Dosing |
Sample size |
Follow-up |
Main Findings |
|
|
Nidorf et al.
2013
|
Randomized, observer-blinded trial. CCS patients were assigned to colchicine or no colchicine |
Primary composite: ACS, out of hospital cardiac arrest, or non- cardioembolic stroke
Secondary: individual components of the primary outcome and the components of ACS unrelated to stent disease |
0.5 mg/day |
Colchicine= 282
Controls= 250 |
3 years |
· Colchicine reduced the primary outcome (5.3% vs 16.0%, HR 0.33, 0.18-0.59, P< 0.001) compared with placebo · Colchicine reduced the incidence of ACS (4.6% vs 13.6%, HR 0.33, 0.18-0.63, P <0.001) and non- cardioembolic stroke (0.35% vs 1.6%, HR 0.23, 0.03-2.03, P =0.011) compared with placebo · In a pre-specified secondary on-treatment analysis that excluded 32 patients assigned to colchicine who withdrew within 30 days due to intestinal intolerance and a further 7 patients (2%) who did not start treatment, the primary outcome occurred less frequently with colchicine compared to placebo (4.5% vs 16.0%, HR 0.29, 0.15-0.56, P< 0.001) |
|
Nidorf et al.
2020
|
Randomized, controlled, double-blind trial. CCS participants were assigned to receive either colchicine or placebo |
Primary composite: cardiovascular death, spontaneous (nonprocedural) MI, ischemic stroke, or ischemia-driven coronary revascularization. Secondary composite: cardiovascular death, spontaneous MI, or ischemic stroke |
0.5 mg/day or no colchicine |
Colchicine= 2762
Controls= 2760 |
28.6 months |
· Colchicine reduced the primary outcome compared with placebo (6.8% vs 9.6%, HR 0.69, 0.57-0.83, P <0.001) · Colchicine reduced the key secondary endpoint (4.2% vs 5.7%, HR 0.72, 0.57-0.92, P =0.007) compared with placebo |
|
Tardif et al.
2019
|
Randomized, double-blind trial involving patients recruited within 30 days after a MI. The patients were randomly assigned to receive either low-dose colchicine or placebo |
Primary composite: death from cardiovascular causes, resuscitated cardiac arrest, MI, stroke, or urgent hospitalization for angina leading to coronary revascularization. Secondary: consisted of the components of the primary end point; a composite of death from cardiovascular causes, resuscitated cardiac arrest, myocardial infarction, or stroke; and total mortality in time-to-event analyses |
0.5 mg/day or placebo |
Colchicine=2366 Placebo= 2379 |
22.6 months |
· Colchicine significantly reduced the primary endpoint (5.5% vs 7.1%, HR 0.77, 0.61-0.96, P =0.02) compared to placebo · Colchicine reduced stroke (HR 0.26, 0.10-0.70), and urgent hospitalization for angina leading to coronary revascularization (HR 0.50, 0.31-0.81), compared to placebo · No difference in the secondary composite endpoints between groups (4.7% vs 5.5%) · No difference in total mortality between groups (1.8% vs 1.8%) · No difference in the incidence of diarrhea between groups (9.7% vs 8.9%, P =0.35) · Increased risk of pneumonia with colchicine compared to placebo (0.9% vs 0.4%, P = 0.03) |
|
Tong et al.
2020
|
Multicenter, randomized, double-blind, placebo-controlled trial. Patients who presented with ACS and had evidence of coronary artery disease on coronary angiography managed with either PCI or medical therapy were assigned to receive either colchicine or placebo
|
Primary composite: all-cause mortality, ACS, ischemia-driven (unplanned) urgent revascularization, and non-cardioembolic ischemic stroke in a time to event analysis
|
0.5 mg twice daily for the first month, then 0.5 mg daily for 11 months or placebo |
Colchicine= 396 Placebo= 399 |
12 months |
· No difference in the primary outcome between groups (24 vs 38 events, P =0.09) · Increase in total death (8 vs 1, P =0.017) and non-CV death (5 vs 0, P =0.024) with colchicine · No difference in adverse effects between groups (23.0% vs 24.3%). The majority of them were gastrointestinal symptoms (23.0% vs 20.8%) |
|
Deftereos et al.
2013
|
Double-blind, prospective, placebo-controlled study. Diabetic patients with contraindication to a drug-eluting stent, undergoing PCI with a BMS, were randomized to colchicine or placebo. Angiography and intravascular ultrasound were performed 6 months after the index PCI |
Primary: Angiographic and IVUS restenosis
Secondary: angiographic and IVUS parameters of lumen loss and in-stent neointimal hyperplasia |
0.5 mg twice daily or placebo |
Colchicine= 100 Placebo=110 |
6 months |
· Colchicine significantly reduced angiographic restenosis (16% vs 33%, P =0.007) and IVUS restenosis (24% vs 43%, P =0.006) · Colchicine significantly reduced minimum lumen diameter (2.8 mm (2.2–3.1) vs 2.3 mm (1.3-2.9), P <0.01) · Gastrointestinal symptoms (diarrhea and nausea): were the most common adverse events in the colchicine group (16% vs 7%, P =0.058) |
|
Opstal et al.
2023
|
Randomized, parallel, double-blind trial that evaluated the effect of adding colchicine or placebo in patients with chronic coronary disease |
Cause-specific mortality data were analysed, stratified by treatment status |
0.5 mg once daily or placebo |
Colchicine=2762 Placebo=2760 |
29 months |
· No difference in CV death between groups (0.7% vs 0.9%)
· No difference in non-CV death between groups (1.9% vs 1.3%)
|
|
Jolly et al.
2024
|
Multicenter trial with a 2-by-2 factorial design. It was randomly assigned patients who had myocardial infarction to receive either colchicine or placebo and either spironolactone or placebo. |
Primary composite: death from cardiovascular causes, recurrent MI, stroke, or unplanned ischemia-driven coronary revascularization |
For the first 90 days: patients weighing >70 kg 0.5 mg x 2/day, if <. 70 kg 0.5 mg /day. After 90 days 0.5 mg /day for all patients |
Colchicine= 3528 Placebo= 3534 |
2.98 years |
· No difference in the primary-outcome between groups (9.1% vs 9.3%, P =0.93)
· Colchicine increased the incidence of diarrhea (10.2% vs 6.6%, P <0.001)
|
|
Mewton et al.
2021
|
Double-blind multicenter trial. Patients admitted for a first episode of STEMI referred for PCI were randomized to receive colchicine or placebo from admission to day 5. Patients underwent a cardiac magnetic resonance at 5 days and at 30 days |
Primary: reduction of IS at 5 days.
Secondary: LV end-diastolic volume change at 3 months and IS at 3 months |
2 mg loading dose followed by 0.5 mg twice a day for 5 days |
Colchicine= 101 Placebo= 91 |
3 months |
· At 5 days IS did not differ between the colchicine and placebo groups (26 g vs 28.4 g of LV mass, P = 0.87) · At 3 months follow-up, there was no significant difference in LV remodeling (colchicine +2.4% vs –1.1% change in LV end-diastolic volume, P =0.49) · Infarct size at 3 months was also not significantly different between the colchicine and placebo groups (17g vs 18 g, P =0.92) · The incidence of gastrointestinal adverse events during the treatment period was greater with colchicine than with placebo (34% vs 11%, P =0.0002) |
|
Bouleti et al.
2023
|
Follow-up analysis of the COVERT-MI study on prespecified secondary clinical endpoints |
Primary composite: all-cause death, ACS, heart failure events, ischaemic strokes, sustained ventricular arrhythmias and acute kidney injury |
2 mg loading dose followed by 0.5 mg two times a day for 5 days or placebo |
Colchicine= 101 Placebo=91 |
1 year |
· No significant difference regarding the number of MACEs between groups · No differences in the occurrence of ischemic strokes: colchicine 3% vs placebo 2.2% (P =0.99) |
|
Deftereos et al.
2015
|
Prospective, double-blinded, placebo-controlled study. Patients presenting with STEMI ≤12 hours from pain onset (treated with PCI) were randomly assigned to colchicine or placebo for 5 days. A subset of patients underwent cardiac MRI 6 to 9 days after the index STEMI (MRI subgroup) |
Primary: area under the curve of CK-MB fraction concentration over 72 hours after admission
Secondary: Maximal high-sensitivity troponin T measure during the same time period.
In MRI subgroup absolute MI volume, determined by LGE, was the primary outcome measure |
Loading dose of 2 mg (1.5 mg initially followed by 0.5 mg 1 hour later) and continuing with 0.5 mg twice daily, or placebo, for 5 days |
Colchicine= 77 Placebo=74 MRI subgroup= 60
|
5 days, until 9 days for MRI subgroup |
· The area under the creatine kinase-myocardial brain fraction curve was for colchicine group 3144 ng·h–1·mL–1 vs placebo group: 6184 ng·h–1·mL–1, P <0.001 · Median maximum high-sensitivity troponin T was 19763 pg/mL and 45 550 pg/mL in the colchicine and control group, respectively (P =0.001) · Indexed MRI-late gadolinium enhancement–defined infarct size was 18.3 mL/1.73 m2 in the colchicine vs 23.2 mL/1.73 m2 in control group (P =0.019) · The relative infarct size (as a proportion to left ventricular myocardial volume) was 13.0 % in the colchicine and 19.8 %, in the control group (P =0.034) |
|
Shah et al.
2020
|
Randomized, double-blind, placebo-controlled trial. Subjects referred for possible PCI were randomized to acute preprocedural oral administration of colchicine or placebo |
Primary: PCI-related myocardial injury according to the Universal Definition
Secondary: Occurrence of 30-day MACEs (earliest occurrence of death from any cause, nonfatal MI, or target vessel revascularization) PCI-related MI as defined by the SCAI (76)
|
1.2 mg 1 to 2 hours before coronary angiography, followed by colchicine 0.6 mg 1 hour later or immediately preprocedure |
Colchicine= 366 Placebo=348
Colchicine+PCI= 206 Placebo+PCI= 194 |
30 days |
· Primary outcome did not differ between colchicine and placebo groups (57.3% vs 64.2%, P =0.19) · Secondary composite outcome: colchicine 11.7% vs placebo 12.9% (P =0.82) · PCI-related MI defined by the SCAI (colchicine 2.9% vs placebo 4.7%, P =0.49) |
|
Cole et al.
2021
|
Randomized pilot trial. Patients undergoing PCI for stable angina or NSTEMI were randomized to oral colchicine or placebo, 6 to 24 hours pre-procedure |
Primary: periprocedural myocardial infarction |
1 mg followed by 0.5 mg 1 hour later |
Colchicine= 36 Placebo= 39
|
24 h |
· No patients developed periprocedural myocardial infarction in either group · Colchicine significantly reduced major periprocedural myocardial -injury: colchicine 31% vs placebo 54%, P =0.04 · Colchicine significantly reduced minor periprocedural myocardial -injury: colchicine 58% vs placebo 85%, P =0.01 |
|
Yu et al.
2024
|
Prospective, single-center, randomized, double-blind clinical trial. Patients with ACS with lipid-rich plaque detected by optical coherence tomography were included. The subjects were randomly assigned to receive either colchicine or placebo |
Primary: Change in the minimal fibrous cap thickness from baseline to the 12-month follow-up
|
0.5 mg once daily |
Colchicine= 52 Placebo=52
|
12 months |
· Colchicine increased the minimal fibrous cap thickness (51.9 μm vs 87.2 μm, P =0.006), and reduced average lipid arc (–25.2° vs –35.7°, P =0.004), mean angular extension of macrophages (–8.9° vs –14.0°, P =0.044) · Colchicine significantly reduced Hs-CRP levels (geometric mean, 0.6 vs 0.3, P =0.046), interleukin-6 levels (geometric mean ratio, 0.8 vs 0.5, P =0.025), and myeloperoxidase levels (geometric mean ratio, 1.0 vs 0.8, Pb =0.047)
|
|
Vaidya et al.
2018
|
Prospective non-randomized observational study. Patients with recent ACS (<1 month), received either colchicine plus OMT or OMT alone |
Primary: change in LAPV a marker of plaque instability on CCTA and robust predictor of adverse cardiovascular events.
Secondary: changes in other CCTA measures and in hs-CRP |
0.5 mg/day plus OMT or OMT alone |
Colchicine= 40 Placebo=40
|
12.6 months |
· Colchicine therapy significantly reduced LAPV (mean 15.9 mm3 [−40.9%] vs 6.6 mm3 [−17.0%], P =0.008) and hs-CRP (mean 1.10 mg/l [−37.3%] vs 0.38 mg/l [−14.6%], P <0.001) vs controls · Atheroma volume (mean 42.3 mm3 vs 26.4 mm3; P =0.28) and low-density lipoprotein levels (mean 0.44 mmol/l vs 0.49 mmol/l, P =0.21) were similar between groups · With multivariate linear regression, colchicine therapy remained significantly associated with greater reduction in LAPV (P =0.039) and hs-CRP (P =0.004) · Significant linear association (P <0.001) and strong positive correlation (r =0.578) between change in LAPV and hs-CRP
|
|
Zuriaga et al.
2024
|
TET2-mutant clonal haematopoiesis was modeled in mice using bone marrow transplants in Ldlr−/− mice, treated with colchicine or placebo. In humans, data from two large biobanks were analyzed to assess if colchicine reduces the link between TET2 mutations and myocardial infarction |
|
In mice: starting with 0.05 mg/kg/day for the first week, and transitioning to 0.1 mg/kg/day for the second week, and 0.2 mg/kg/day for the remaining 6 weeks |
Humans Colchicine=3849 Non cholchicine users: 433387
|
- |
Mouse Model · Colchicine reduced plaque size by ~27% in TET2-KO mice, P =0.003 No effect on WT controls, P =0.693 · Colchicine reduced IL-1β production in TET2-KO macrophages more significantly than WT macrophages (48% vs 16% decrease, P = 0.005) · IL-6 reduction was similar in both genotypes (~70%)
Human Studies · Colchicine use attenuated the risk of MI (OR colchicine 0.76, 0.43–1.34 vs OR no colchicine 1.23 (0.90-1.56, Pint= 0.04) · Colchicine reduced MI (HR colchicine 0.30, 0.08–1.22 vs HR no colchicine 1.08, 0.93-1.10, Pint= 0.05)
|
Abbreviations: CCS= Chronic Cornary Syndrome; ACS= Acute Coronary Syndrome; MI= Myocardial Infarction; PCI=Percoutaneous Coronary Intervention; BMS=Bare-metal Stent; IVUS=Intravascular Ultrasound; CV= Cardiovascular; STEMI =ST-Elevation Myocardial Infarction; IS=Infarct Size; LV=Left Ventricle; MACEs= Major Adverse Cardiovascular Events; CK-MB=Creatine Kinase-Myocardial Brain; MRI=Magnetic Resonance Imaging; LGE= Late Gadolinium Enhancement; SCAI=Society for Cardiovascular Angiography and Interventions; NSTEMI= Non ST-Elevation Myocardial Infarction; Hs-CRP= High Sensitivity C-reactive Protein; OMT=Optimal Medical Therapy; LAPV =Low Attenuation Plaque Volume; CCTA= Coronary Computed Tomography Angiography; KO=Knock Out; WT=Wild Type.
- Address the bleeding risk associated with dual antiplatelet therapy and dual-pathway inhibition in more detail. Consider stratifying patients based on bleeding risk to contextualize the strategies better.
Author Reply: As mentioned below, the primary aim of this study is to summarize the available clinical and pre-clinical evidence on the combined use of colchicine with antiplatelet therapy to address atherothrombosis. We have now better specified the aim of the manuscript in the abstract and in the introduction section as above mentioned. Additionally, we contextualized this strategy within the framework of other currently recommended approaches, such as dual antiplatelet therapy and dual-pathway inhibition. However, no evidence on the comparative effects of a strategy of colchicine plus antiplatelet therapy vs DAPT or DPI are currently available. Therefore, any argumentation on this regard would not be supported by evidence, resulting in a mere speculation. Moreover, such a discussion would go beyond the scope of this analysis, making it difficult to fall within the word count allowed by the journal. Accordingly, as per the suggestion by other Reviewers, we have removed the former table 1.
However, in the attempt to at least partially comply with the comment by this Reviewer, we have now added the following sentence at page 20: “Finally, evidence comparing the safety and efficacy of a strategy combining colchicine with antiplatelet agents versus DAPT or DPI, alongside studies aimed at identifying the specific patient profiles most likely to benefit from dual targeting of inflammatory and platelet pathways rather than focusing solely on thrombotic pathways, could pave the way for personalized treatment approaches and better clinical outcomes.”.
- Provide more practical insights for clinicians. For instance, offer guidance on how to select between dual antiplatelet therapy, dual-pathway inhibition, and colchicine for specific patient populations.
Author Reply: please refer to the previous comment.
